# Fast gradient-free optimization of excitations in variational quantum eigensolvers
Jonas Jäger [1,2,3,4 ✉], Thierry N. Kaldenbach [1 ✉], Max Haas [1,5] & Erik Schultheis [1,5]

Finding molecular ground states and energies with variational quantum eigensolvers is central to chemistry applications on quantum computers. Physically motivated ansätze based on excitation operators respect physical symmetries, but existing quantum-aware optimizers, such as Rotosolve, have been limited to simpler operator types. To fill this gap, we introduce ExcitationSolve, a fast quantum-aware optimizer that is globally-informed, gradient-free, and hyperparameter-free. ExcitationSolve extends these optimizers to parameterized unitaries with generators $G$ of the form $G^3 = G$ exhibited by excitation operators in approaches such as unitary coupled cluster. ExcitationSolve determines the global optimum along each variational parameter using the same quantum resources that gradient-based optimizers require for one update step. We provide optimization strategies for both fixed and adaptive variational ansätze, along with generalizations for simultaneously selecting and optimizing multiple excitations. On molecular ground state energy benchmarks, ExcitationSolve outperforms state-of-the-art optimizers by converging faster, achieving chemical accuracy for equilibrium geometries in a single parameter sweep, yielding shallower adaptive ansätze and remaining robust to real hardware noise. By uniting physical insight with efficient optimization, ExcitationSolve paves the way for scalable quantum chemistry calculations.

The choice of ansatz for the parameterized or variational quantum circuit plays a crucial role in the variational quantum eigensolver (VQE)[1], which aims to prepare the ground state of a Hamiltonian and find the corresponding ground state energy. The Hamiltonian can describe, e.g., an electronic structure problem in a molecule or material[1–4]. Physically-motivated ansätze, such as a composition of excitation operators like single and double fermionic excitations in the *Unitary Coupled Cluster* (UCCSD) ansatz[1], are particularly relevant because of their guarantees of producing physically plausible states. By design, relevant physical properties of an initial reference state, typically the Hartree-Fock (HF) state, are conserved, such as the number of electrons or spin symmetries. Furthermore, number-conserving, yet hardware-efficient approaches, such as *qubit-excitation based* (QEB) ansätze[5] exist, most prominently appearing in the *Qubit Coupled Cluster Singles Doubles* (QCCSD) ansatz[6]. In contrast, problem-agnostic ansätze like generic hardware-efficient ansätze might yield physically implausible states and energies by, e.g., not conserving the number of particles[7]. These ansätze are composed of parameterized qubit rotations. The implications of ansatz choice are visualized in Fig. 1a.

After specifying the variational ansatz $U(\boldsymbol{\theta})$, its $N$ parameters $\boldsymbol{\theta} \in (-\pi, \pi]^N$ have to be optimized to prepare the desired ground state. This optimization happens iteratively in a hybrid loop involving a quantum computer to evaluate the energy and a classical computer to optimize the parameters. On the quantum computer we evaluate the expectation value $\langle \psi(\boldsymbol{\theta})|H|\psi(\boldsymbol{\theta})\rangle$ of the Hamiltonian $H$ with respect to prepared $n$-qubit state $|\psi(\boldsymbol{\theta})\rangle = U(\boldsymbol{\theta})|\psi_0\rangle$, as a function of the current parameters. The energy landscape $f(\boldsymbol{\theta})$ we want to minimize can be written as

$$f(\boldsymbol{\theta}) = \langle H \rangle = \langle \psi_0|U^\dagger(\boldsymbol{\theta})HU(\boldsymbol{\theta})|\psi_0\rangle. \quad (1)$$

However, the VQE optimization problem is generally challenging because the energy landscape is a $N$-dimensional trigonometric function[8], leading to a large number of local minima, many of which are sub optimal[9–12]. Therefore, gradient-based optimizers (e.g., gradient descent, Adam[13] or BFGS[14–17]) as well as a gradient-free black-box optimizers (e.g., COBYLA[18], SPSA[19,20]) struggle to navigate the complex energy landscape as for larger molecules or materials. Note that, instead of a fixed ansatz $U$, an adaptive

[1]German Aerospace Center (DLR), Institute of Materials Research, Cologne, Germany. [2]University of British Columbia (UBC), Department of Computer Science, Vancouver, BC, Canada. [3]University of British Columbia (UBC), Institute of Applied Mathematics, Vancouver, BC, Canada. [4]Stewart Blusson Quantum Matter Institute, Vancouver, BC, Canada. [5]These authors contributed equally: Max Haas, Erik Schultheis. ✉e-mail: jojaeger@cs.ubc.ca; thierry.kaldenbach@dlr.de

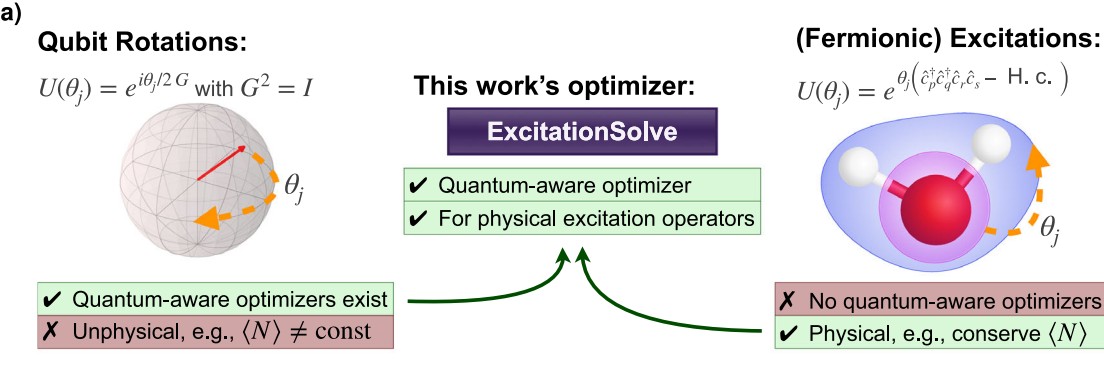

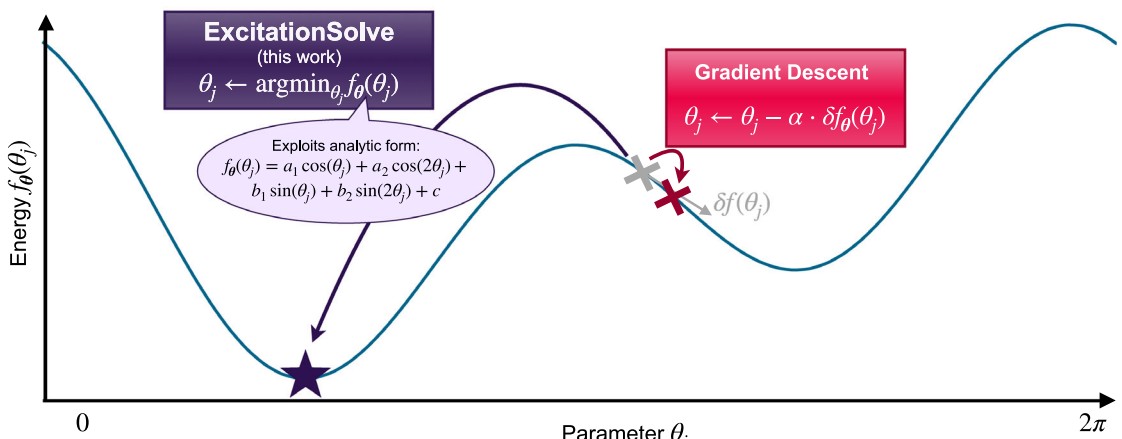

**Fig. 1 | Schematic overview of our work. a** While hardware-efficient ansätze, typically composed of parameterized rotations, allow for fast quantum-aware optimization[8,22 -- 24], they do not preserve physical properties[7], e.g., vary the average particle number $\langle N \rangle$, the opposite is true for physically-motivated ansätze such as those assembled from fermionic excitation operators[1]. Our new optimizer, *ExcitationSolve*, fills this gap and combines fast optimization with physical guarantees. **b** ExcitationSolve (purple) relies on the same quantum resources, i.e., same number of energy measurements, to jump to the global energy minimum along a single parameter $\theta_j$, as a gradient-based optimizer (red) evaluating and following the (partial) derivative in $\theta_j$. The latter does not consider global information of the energy landscape, thus being limited to a local parameter region. Note that since gradient descent is based on the full gradient evaluated over $N$ parameters, ExcitationSolve in fact performs $N$ update steps while gradient descent updates locally once.

ansatz can also be employed, where operators are iteratively added to the ansatz during the optimization. For VQE, this concept was first introduced as ADAPT-VQE[21].

*Quantum-aware* optimizers, defined as those that leverage problem-specific knowledge and properties of the quantum system to be optimized, pose a promising alternative. In contrast to optimizers that treat the quantum system as a *black box* (e.g., the ones listed above), they utilize this information to navigate the energy landscape more efficiently. A prominent example is the *Rotosolve* optimization method[22], which was simultaneously proposed under the term *Sequential Minimal Optimization* (SMO)[23], as well as analogously mentioned in refs. [8, 24]. Rotosolve globally optimizes parameterized operators individually based on the closed form of the energy landscape slice, thus leveraging the operator-specific properties to significantly reduce the required quantum resources[22]. This provides an efficient alternative to gradient-based optimization. However, the type of parameterized operators or gates incorporated in the ansatz must be compatible with the quantum-aware optimizer. The applicability of Rotosolve is limited to unitaries with self-inverse generators, e.g., (Pauli) rotation gates, although generalizations were suggested[25]. While the more complicated unitaries relevant for quantum chemistry applications can be decomposed into fixed entangling gates and parameterized rotations[26], Rotosolve's performance degrades as it then overestimates the required number of energy evaluations. While these gradient-free optimizers, like their gradient-based counterparts, are generally only guaranteed to converge to a local optimum,

empirical evidence demonstrates their effectiveness across various applications[8,22–24]. Variations of these optimizers, such as Free-Axis or Free-Quaternion Selection[27–31] and the Unitary Block Optimization Scheme[32] further support their utility.

In this work, we combine the advantages of incorporating excitation operators in physically-motivated VQE ansätze with the effectiveness of quantum-aware optimizers. We introduce *ExcitationSolve*, a fast globally-informed gradient-free optimizer for ansätze composed of excitation operators. ExcitationSolve is quantum-aware since we know the analytical form of the energy landscape in one parameter (or a subset of parameters) for excitation operators, which is a (multi-dimensional) *second-order Fourier series*. This applies to fermionic excitations[1], qubit excitations[5,6], and Givens rotations[33]—not limited to single and double excitations. ExcitationSolve can be applied to fixed and adaptive VQE ansätze as in the UCCSD ansatz[1] and ADAPT-VQE[21], respectively. Within the literature on such optimizers, ExcitationSolve can be characterized as an extension of Rotosolve[8,22–24] and Greedy Gradient-free Adaptive VQE (GGA-VQE)[34] for VQE and ADAPT-VQE, respectively. A schematic summary of our work is provided in Fig. 1.

## Results

### ExcitationSolve algorithm
In this section, we introduce the quantum-aware optimization algorithm *ExcitationSolve*, which readily extends Rotosolve-type optimizers[22,23] to

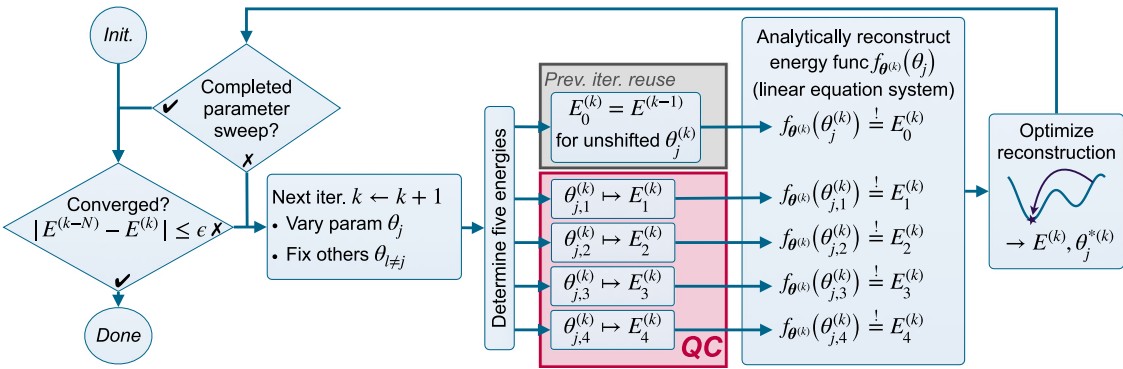

**Fig. 2 | Flow chart for ExcitationSolve for fixed ansätze.** In this iterative algorithm, the $k$-th iteration updates a single parameter $\theta_j$ through repeated sweeps over all $N$ parameters until convergence. To reflect the flexibility in the sweep order, we use separate indices $j$ and $k$. Per iteration, the parameter is shifted to four different positions $\theta_{j,1}^{(k)}, \theta_{j,2}^{(k)}, \theta_{j,3}^{(k)}, \theta_{j,4}^{(k)}$, and the quantum computer (QC) is used to obtain the corresponding energy values. This is the only part requiring the quantum hardware (purple). All remaining steps are efficiently computed classically. The energy associated with the unshifted current parameter value $\theta_j^{(k)}$ is re-used from the previous iteration $k-1$.

excitation operators, which obey the following more general form. Throughout this work, we assume variational ansätze $U(\boldsymbol{\theta})$ consisting of a product of unitary operators $U(\theta_j)$ of the generic form

$$U(\theta_j) = \exp(-i\theta_j G_j), \qquad (2)$$

depending on a single parameter $\theta_j$ each (the $j$-th component of $\boldsymbol{\theta}$). Most importantly, with the Hermitian generators $G_j$ fulfilling $G_j^\dagger = G_j$. Note that any generator with $G_j^2 = I$ (the prerequisite of Rotosolve) fits into this description. However, this work is concerned with the class of excitation operators because their generators fulfill $G_j^3 = G_j$ and, importantly, $G_j^2 \neq I$.

In the following, we first present the analytic form of the energy landscape when varying a single parameter in an operator of the aforementioned structure, and, second, how this is exploited to derive an optimization algorithm. The analytic form of the energy with respect to a single parameter $\theta_j$ associated with some generator $G_j$ is a finite Fourier series (also known as a trigonometric polynomial) of second-order with period $2\pi$ and has the form

$$f_{\boldsymbol{\theta}}(\theta_j) = a_1 \cos(\theta_j) + a_2 \cos(2\theta_j) + b_1 \sin(\theta_j) + b_2 \sin(2\theta_j) + c. \qquad (3)$$

Here, the notation $f_{\boldsymbol{\theta}}(\theta_j)$ refers to the energy landscape $f(\boldsymbol{\theta})$ from Eq. (1) with all parameters being fixed except $\theta_j$. The five coefficients $a_1, a_2, b_1, b_2, c$ are independent of the parameter $\theta_j$ but may depend on the remaining parameters $\theta_{i\neq j}$, which is detailed in the constructive proof in Supplementary Note 4.1. In order to determine these five coefficients, we need energy values in at least five distinct configurations of the parameter $\theta_j$. For five evaluations, the coefficients are the solution to the linear equation system, whereas for more than five evaluations, the overdetermined equation system can be solved using either the least square method or truncated (fast) Fourier transform. Supplementary Note 3.4 discusses how this relates to noise robustness.

The proposed optimization algorithm ExcitationSolve (cf. Fig. 2) iteratively sweeps through the $N$ parameters $\boldsymbol{\theta}$, reconstructs the energy landscape per parameter $\theta_j$ analytically, globally minimizes the reconstructed function classically, and assigns the parameter $\theta_j$ to the value where the global minimum is attained. Hence, each parameter sweep consists of $N$ updates, and the order in which the $N$ parameters are optimized can be chosen freely. This process is repeated until convergence, which is defined by a threshold criterion on the absolute or relative energy reduction of the last parameter sweep. In this algorithm, the quantum computer is used exclusively to obtain energy evaluations while the reconstruction and subsequent minimization of the energy landscape is performed on a classical computer. To determine the minimum energy and corresponding parameter classically, we utilize a companion-matrix method[35], which is a direct

numerical method detailed in the Methods. Importantly, in each optimization step, the previously determined minimum energy can be reused, requiring only an additional *four* parameter shifts to reconstruct the energy landscape along the next parameter. Supplementary Note 3.1 describes the exact algorithmic details. It should be emphasized that for the specific analytic form in Eq. (3) each parameter $\theta_j$ must occur only once in the ansatz in terms of parameterized excitations, which is a commonly satisfied assumption. This number of occurrences should not be confused with the number of single-qubit rotations arising when further decomposing an excitation into basic gates, which in general will be more than one. We yet further generalize ExcitationSolve to multiple occurrences of the same parameter in the Methods, e.g., making it compatible with ansätze constituted of higher-order product formulas or multiple Trotter steps.

In essence, ExcitationSolve performs a gradient-free *coordinate descent* with (efficient) *exact line search*, i.e., independently optimizing each parameter $\theta_j$ iteratively until convergence. For the exact line search, it leverages an efficient analytic reconstruction of the energy landscape in a *single* parameter to determine its *global* optimum directly while the other parameters $\theta_{i\neq j}$ remain fixed. Most importantly, the effective resource demands on the quantum hardware per parameter are equivalent to gradient-based optimizers.

**Supported types of excitation operators.** Excitation operators are one class of operators whose generators satisfy $G^3 = G$. Such operators appear for example as generators in UCC theory[2,36], which is a post Hartree-Fock method that unitarily evolves the Hartree-Fock ground state based on fermionic excitations. For a fermionic excitation of $m$ electrons, the $m$-excitation generator reads

$$\tau_{\boldsymbol{o},\boldsymbol{v}}^{(m)} = \prod_{l=1}^{m} a_{v_l}^\dagger a_{o_l} - \text{H.c.}, \qquad (4)$$

where $a^\dagger/a$ are the standard fermionic creation/annihilation operators and the $m$-component vectors $\boldsymbol{o}/\boldsymbol{v}$ entail the involved occupied/virtual orbitals, respectively. The product $\prod_l$ is intuitively taken right-to-left, however, changing this order conveniently leaves the expression invariant. The corresponding $m$-electron unitary excitation operator is defined as

$$U_{\boldsymbol{o},\boldsymbol{v}}^{(m)}(\theta) = \exp\left(\theta \tau_{\boldsymbol{o},\boldsymbol{v}}^{(m)}\right). \qquad (5)$$

To incorporate all eligible $m$-electron excitations from occupied to virtual orbitals, the $m$-th cluster operator $T^{(m)} = \sum_{\boldsymbol{o},\boldsymbol{v}} \theta_{\boldsymbol{o},\boldsymbol{v}} \tau_{\boldsymbol{o},\boldsymbol{v}}^{(m)}$ is introduced. The truncated cluster operator, including all excitations of $M$ or less electrons, is defined as $T = \sum_{m=1}^{M} T^{(m)}$ and serves as the generator of the variational

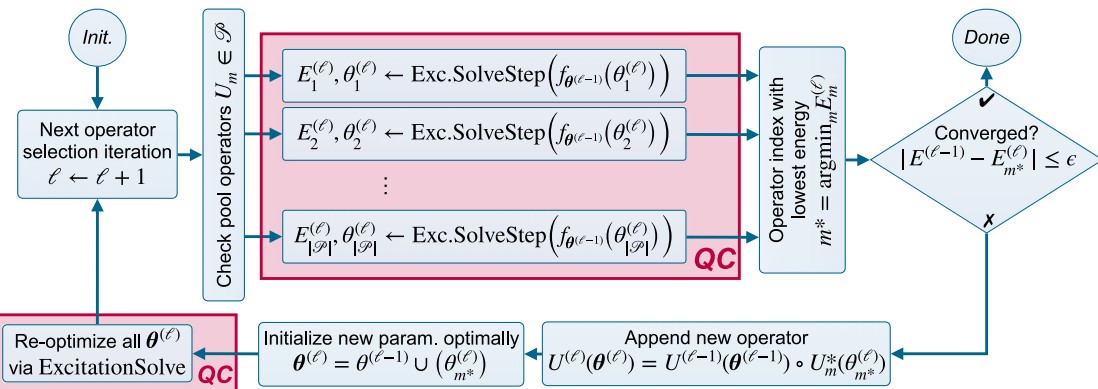

**Fig. 3 | Flow chart for ExcitationSolve for ADAPT-VQE (adpative ansätze).** ExcitationSolve is integrated into ADAPT-VQE in two parts. First, for the selection criterion from the operator pool $\mathscr{P}$ by determining the immediate energy improvements via a single ExcitationSolve iteration when appending each of the operator candidates $U_m$ separately. Second, to re-optimize all parameter $\boldsymbol{\theta}^{(\ell)}$ at the end of each ADAPT iteration. The usage of the quantum computer (QC, red) solely happens in ExcitationSolve(Step) (Fig. 2) when invoked as sub-routines. Note that ADAPT iteration $\ell$ denotes how many operators have been appended to the ansatz.

unitary. Typically, the unitary $\exp(T)$ is then approximated through a first-order Trotter-Suzuki[37] decomposition

$$U(\boldsymbol{\theta}) = \prod_{m=1}^{M} \prod_{\boldsymbol{o},\boldsymbol{v}} U_{\boldsymbol{o},\boldsymbol{v}}^{(m)}(\theta_{\boldsymbol{o},\boldsymbol{v}}), \tag{6}$$

where $\boldsymbol{\theta}$ contains all the variational parameters $\theta_{\boldsymbol{o},\boldsymbol{v}}$. In Supplementary Note 4.2 we show that the anti-Hermitian fermionic excitation operators (Eq. (4)) obey the equation $\tau^3 = -\tau$ for arbitrary excitation-orders. Consequently, we can define the Hermitian generator $G = i\tau$ with $G^3 = G$, such that the excitation operators from Eq. (4) comply with the form in Eq. (2). Thus, the energy landscape $f(\boldsymbol{\theta}) = \langle U^{\dagger}(\boldsymbol{\theta}) H U(\boldsymbol{\theta}) \rangle$ in a single parameter takes the form from Eq. (3).

In practice, the truncated cluster operator is often restricted to only include single-electron- and double-electron excitations ($M = 2$), resulting in the *Unitary Coupled Cluster Singles Doubles* unitary (UCCSD). We note, that ExcitationSolve is applicable for arbitrary truncation orders $M$ and, importantly, that the required energy evaluations for the optimization stays constant regardless of the order of the excitation $m$ the energy always obeys a second-order Fourier series. In contrast, in order to optimize excitation operators using Rotosolve/SMO, one applies a fermionic mapping, e.g., Jordan-Wigner (JW)[38] or Bravyi-Kitaev (BK)[39–41], to decompose the operation into compatible Pauli rotations. The order of the Fourier series predicted by SMO scales *exponentially* in the order of the excitation operator[42,43]. We further emphasize that this approach works for any fermion-to-qubit mapping. Analogously to fermionic excitations, one can use other types of excitations such as *qubit excitations* used in QEB ansätze such as QCCSD[5,6] (recently explored in ref. 34) also sometimes referred to as *Givens rotations*, or *controlled* excitations[33]. The latter are universal for particle-number preserving unitaries. We further note that ExcitationSolve can be readily applied to the recently introduced couple exchange operators (CEOs) consisting of linear combinations of excitations[44]. In particular, the linear combination of two distinct excitations acting on the same (spin-) orbitals with one shared variational parameter (OVP-CEOs) satisfies the requirements. As a final note, our algorithm is readily applicable to generalized excitations[45,46], which do not distinguish between occupied and virtual orbitals.

## ExcitationSolve for ADAPT-VQE: Globally-informed selection.

When optimizing adaptive ansätze, e.g., ADAPT-VQE[21], a scoring criterion is needed to select an operator from the pool to append to the ansatz. The goal of this criterion is to assess the effectiveness of this operator selection in producing the ground state and energy. Naturally, we apply ExcitationSolve to ADAPT-VQE (cf. Fig. 3) to obtain a globally-

informed ADAPT-VQE operator selection criterion by leveraging analytic energy reconstructions for each operator candidate separately when added to the current ansatz. We select the operator that achieves the strongest *immediate* decrease in energy to be appended to the current ansatz (and parameters) and initialize it in its optimal value. Given the potential for a stronger energy decrease by adjusting the preceding parameters, we use ExcitationSolve to re-optimize all parameters in the typical fixed ansatz VQE manner before extending the ansatz further. We only proceed to the next ADAPT(-VQE) iteration if the threshold criterion for convergence has not yet been met. The details are described in Supplementary Note 3.2. Note that a similar approach, known as *Greedy Gradient-free Adaptive VQE (GGA-VQE)*, was recently proposed[34], but it was limited to parameterized qubit excitations and rotations and neglected the re-optimization of the intermediate parameters. In contrast we extend it to fermionic excitation operators and include effective re-optimization. We further note that the energy ranking of the operator pool can be utilized to append the top two (or more) operators at once, as recently proposed in ref. 34. This is only a heuristic for the most impactful pair (or subset) of operators, since the largest individual impact does not necessarily imply the largest simultaneous impact. However, an efficient initialization in their simultaneous optimum can still be achieved via the multi-parameter extension of ExcitationSolve (see the "Multi-parameter generalization" subsection).

Figure 4 demonstrates the advantage of our globally-informed selection criterion by comparing it with the original local ADAPT-VQE criterion[21], which selects operators based on the magnitude of their partial derivative at zero $|f'_{\boldsymbol{\theta}}(0)|$ (details provided in the Supplementary Note 5.3). In contrast, ExcitationSolve assesses the potential impact of each operator on a global scale, identifying the operator that provides the greatest immediate improvement. Note that the example provided in Fig. 4 is fabricated through a random three-qubit Hamiltonian to showcase the motivation and does not correspond to an actual scenario observed in numerical simulations. From a theoretical point of view, a valuable insight can be made by considering that the original ADAPT-VQE criterion approximates the energy landscape in the selected operator's parameter $f_{\boldsymbol{\theta}}(\theta_{N+1})$ by a *first-order* Taylor expansion around $\theta_{N+1} = 0$, while ExcitationSolve utilizes the exact energy landscape or, analogously, the *full* Taylor series, i.e.,

$$f_{\boldsymbol{\theta}}(\theta_{N+1}) = \overbrace{\underbrace{f_{\boldsymbol{\theta}}(0) + \frac{1}{1!}f'_{\boldsymbol{\theta}}(0)\theta_{N+1}}_{\text{Original ADAPT-VQE}} + \frac{1}{2!}f''_{\boldsymbol{\theta}}(0)\theta_{N+1}^2 + \frac{1}{3!}f'''_{\boldsymbol{\theta}}(0)\theta_{N+1}^3 + \cdots}^{\text{ExcitationSolve ADAPT-VQE}}.$$

$$\tag{7}$$

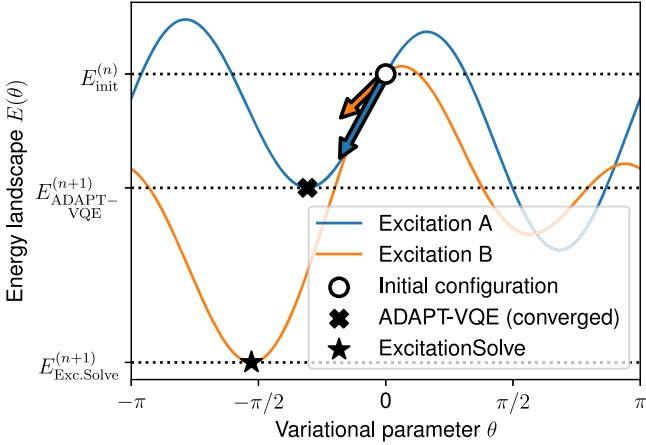

**Fig. 4 | ADAPT-VQE vs. ExcitationSolve.** We consider the selection among two excitation operator candidates A and B in the adaptive setting. In the original ADAPT-VQE approach[21], excitation A is selected based on the gradient criterion, i.e. the steepest gradient at $\theta = 0$. This is then converged with a gradient descent towards a (potentially only local) minimum, typically requiring multiple gradient evaluations. ExcitationSolve chooses excitation B (despite the smaller gradient at $\theta = 0$) based on the energy criterion, i.e., the attainable global energy minimum, and already initializes $\theta$ in its optimal configuration.

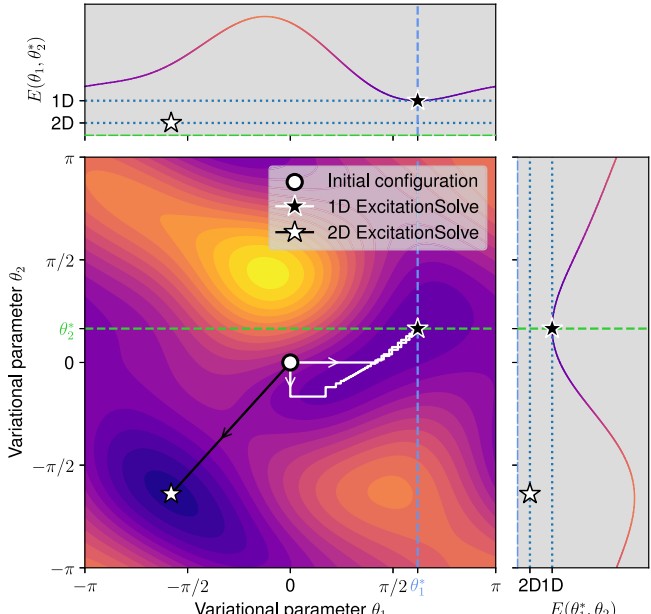

**Fig. 5 | 1D ExcitationSolve with coordinate descent vs. 2D ExcitationSolve.** The simultaneous optimization of two parameters can be achieved either by effectively reducing it to a 1D optimization task using coordinate descent or employing a true 2D optimization based on the energy landscape from Eq. (8). Color indicates energy linearly from low (violet) to high (yellow). In this example, no matter which parameter is tuned first, the 1D coordinate descent approach (white arrows) converges only to a local minimum (black star marker). Also, this convergence takes up multiple iterations. Meanwhile, in the proper 2D case, the global reconstruction of the 2D second-order Fourier series permits an immediate jump (black arrow) to the global minimum (white star marker).

Given that the first-order Taylor approximation is a linear function, the minimum energy is trivially attained at either boundary, $\theta_{N+1} = \pm\pi$, with energy $f_{\boldsymbol{\theta}}(0) \pm f'_{\boldsymbol{\theta}}(0)\pi$. Thus, the original ADAPT-VQE selects the operator from the pool that decreases the energy the most in the first-order Taylor approximation (in $\theta_{N+1} = 0$) of the energy landscape, whereas ExcitationSolve does so based on the full Taylor series, i.e., the exact energy landscape. Especially for such trigonometric functions, a first-order Taylor approximation falls short in faithfully capturing the up to four optima and identifying the global optimum, highlighting the effectiveness of the global criterion in ExcitationSolve. This underscores the importance of such global criteria for operators that induce higher-degree trigonometric functions, unlike rotations corresponding to simple sine curves where local minima trivially coincide with global minima.

**Multi-parameter generalization.** In this section, we generalize the one-dimensional optimization to multiple dimensions, i.e. independent parameters. As illustrated in Fig. 5, this enables ExcitationSolve to potentially avoid and escape local minima in the energy landscape because a improved local or global optimum may be unveiled in a higher-dimensional space. The multi-parameter generalization can be used for both the optimization of fixed and adaptive ansätze. In the context of rotation operators, this has already been explored[23]: The energy varied in $D$ parameters is analytically described by a $D$-dimensional first-order Fourier series. Analogously, we show in Supplementary Note 4.3 that a simultaneous variation of $D$ excitation operators can be described through a $D$-dimensional *second-order* Fourier series. The energy landscape can thus be expressed as

$$f_{\boldsymbol{\theta}}(\boldsymbol{\theta}_{\mathcal{M}}) = \boldsymbol{c} \cdot \left[ \bigotimes_{i \in \mathcal{M}} \begin{pmatrix} \cos(\theta_i) \\ \cos(2\theta_i) \\ \sin(\theta_i) \\ \sin(2\theta_i) \\ 1 \end{pmatrix} \right], \quad (8)$$

where $\boldsymbol{c}$ is a $5^D$-dimensional real-valued vector and $\mathcal{M}$ denotes the index set of the $|\mathcal{M}| = D$ simultaneously varied parameters. Consequently, the full reconstruction of the energy landscape requires a total of $5^D - 1$ new energy evaluations. Once reconstructed, we can classically find the

minimum of the energy landscape, our method of choice is detailed in the Methods.

The exponential number of energy evaluations in the number of parameters hinders multi-parameter ExcitationSolve from being always blindly employed. Nonetheless, it offers a useful tool when employed in the right place. Figure 5 demonstrates an example of when a single application of a 2D optimization not only requires significantly fewer resources than the individual 1D optimization to converge, but also finds the global minimum instead of a local one. Again, this specific example is fabricated through a random three-qubit Hamiltonian. Generally, it can possibly set the optimizer on a more profitable path any time the 1D optimization reaches a local minimum that it cannot escape. To complete the scope of applicability of ExcitationSolve, Supplementary Note 3.3 covers the most generic case of a multi-parameter optimization where each parameter may occur multiple times.

**Experiments**

In this section, we assess the performance of ExcitationSolve on both fixed and adaptive ansätze. We compare it to other optimizers commonly found in VQE literature: Gradient Descent (GD), Constrained Optimization By Linear Approximation (COBYLA)[18], Adam[13], Simultaneous Perturbation Stochastic Approximation (SPSA)[19,20] and the Broyden-Fletcher-Goldfarb-Shannon (BFGS) algorithm[14–17]. The VQE is initialized with the Hartree-Fock (HF) state and parameters set to zero such that the initial energy is the HF energy $E_{\mathrm{HF}}$. To evaluate the experiments, we consider the absolute error between the VQE energy $E_{\mathrm{VQE}}$ with respect to the exact Full Configuration Interaction (FCI) energy $E_{\mathrm{FCI}}$, i.e., $|E_{\mathrm{VQE}} - E_{\mathrm{FCI}}|$. This approach helps us determine when the error falls below the desirable *chemical accuracy* of $10^{-3}$ Ha[1]. Note that, although the term chemical accuracy is commonly used in quantum computing literature, it should more precisely be referred to as chemical precision[2,47].

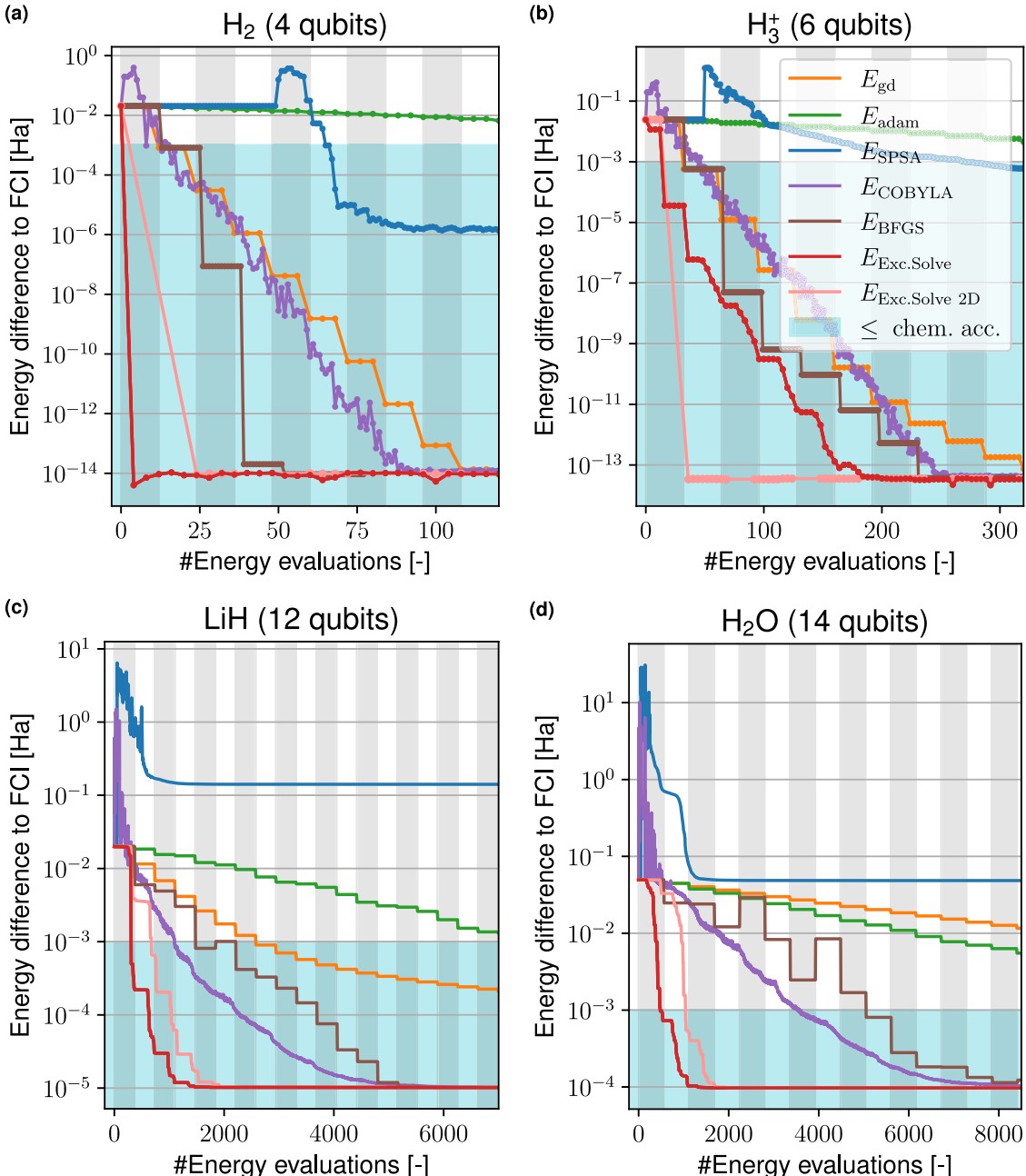

**Fig. 6 | Comparison of optimizers for fixed UCCSD ansätze.** The optimizers under consideration are ExcitationSolve (red), COBYLA (purple), Gradient descent (yellow), Adam (green), SPSA (blue) and BFGS (brown). The plots show the error of the VQE with respect to the Full Configuration Interaction (FCI) solution $|E_{VQE} - E_{FCI}|$ over the number of energy evaluations for the molecules **a** $H_2$, **b** $H_3^+$, **c** LiH and **d** $H_2O$ in their respective equilibrium geometries. The light blue region signifies the chemical accuracy ($10^{-3}$ Ha), while alternating vertical shadings mark full sweeps over all parameters. As each sweep incurs the same cost as a single gradient computation via the parameter-shift rule, gradient-based optimizers exhibit piecewise constant progress in comparison. The BFGS optimizer needs additional energy evaluations per update step to approximate the Hessian. Therefore, the BFGS updates do not align with the vertical shadings.

The quantum resource demand of the optimizers is tracked in the number of *energy evaluations*, which refers to obtaining the expectation value of the Hamiltonian and is proportional to the actual number of measurements and terms of the Hamiltonian. Gradients for GD and Adam are computed using the four-term parameter-shift rule, requiring four energy evaluations per partial derivative. This equivalence in terms of energy evaluations provides a common basis for directly comparing all optimizers. By reusing previous energies, ExcitationSolve matches the cost of one iteration (parameter sweep) to that of one partial derivative (gradient) computation. The presented results are for optimizers with tuned hyperparameters. Full details on the experiments, their implementation and evaluation can be found in Supplementary Note 1.

**Fixed ansatz (UCCSD) comparison with other optimizers.** To compare the optimizers we use a fixed ansatz where the tunable parameters and the order of the parameters are exactly the same for all optimizers. We choose the UCCSD ansatz in its first-order Trotter-approximation where we apply a single layer of first all double and then all single excitations. Figure 6 presents the results where we studied the ground state energies of the molecules $H_2$ (4 qubits, Fig. 6a), $H_3^+$ (6 qubits, Fig. 6b),

LiH (12 qubits, Fig. 6c) and $H_2O$ (14 qubits, Fig. 6d), each in their equilibrium geometry[48] in the STO-3G basis.

We find that ExcitationSolve does not only take fewer evaluations to reach chemical accuracy but also achieves this within a single sweep over the parameters. ExcitationSolve finds the exact ground state energy faster than all other optimizers with the most prominent speedup for larger molecules. For the $H_2$ molecule, ExcitationSolve even converges to the FCI energy in just one VQE iteration. This efficiency is attributed to the ground state being a superposition of the Hartree-Fock state and one doubly-excited state. Therefore, optimizing only the parameter in the UCCSD ansatz corresponding to the relevant double excitation is sufficient for convergence, which ExcitationSolve accomplishes optimally in a single VQE iteration. Similarly, for the $H_3^+$ molecule, ExcitationSolve with a 2D optimization strategy converges to the FCI energy after one 2D optimization, since the FCI ground state is a superposition of the HF state and two excited states. When using ExcitationSolve for both $H_2$ and $H_3^+$ we note energy increases that appear around 5 and 170 energy evaluations, respectively. We attribute these increases in energy to numerical inaccuracies since the energy differences to the exact FCI energy are below $10^{-13}$ Ha in both cases. For $H_2O$ ExcitationSolve reaches both the chemical accuracy and exact solution 7 times faster than the next best optimizer (COBYLA and BFGS, respectively). In contrast, the slowest, yet converging, optimizer (GD) takes 46 and 86 times longer to reach these accuracies. It is worth noting for larger molecules that all optimizers consistently converge with a higher error, likely due to the absence of higher-order excitations in the UCCSD ansatz and the limited expressivity of a single first-order Trotter step. ExcitationSolve readily applies to higher-order excitations, and the "ExcitationSolve for multiple occurrences of parameters" subsection in the Methods provides an extension to ansätze with repeated parameters resulting from higher-order Trotterization. As both SPSA and Adam have a very slow convergence and in some cases do not even manage to reach the chemical accuracy, we disregard them for further studies.

**ADAPT-VQE**. As often suggested in recent literature on variational algorithms[49], fixed ansätze may not be the way forward. We therefore probe ExcitationSolve in an adaptive setting where not only the parameters are optimized using ExcitationSolve, but also the choice of the next operator to append to the ansatz is made using the same strategy. We compare it to the original ADAPT-VQE implementation[21] where the operator selection is made by the gradient criterion and the re-optimization is performed with GD. Both are initialized in the HF state.

Figure 7 shows the convergence of the adaptive optimizations to the ground states of molecules $H_2$, $H_3^+$, LiH and $H_2O$ in their equilibrium geometry. We compare ADAPT-VQE with GD as optimizer to ExcitationSolve and to a 2D variant of ExcitationSolve. Here in each adapt step not only one, but the two best operators are selected, 2D optimized with respect to both parameters $\theta_i$, $\theta_j$ and then appended to the ansatz. All further optimization is performed using 1D ExcitationSolve. The total number of evaluations is composed of the evaluations to select a new operator and the evaluations to re-optimize the parameters already present in the ansatz. The former lead to plateaus, during which the energy remains unchanged. For all four molecules ExcitationSolve reaches faster convergence than ADAPT-VQE for both the chemical accuracy and the limit within the UCCSD ansatz. The reason for the faster convergence stems mainly from two key advantages that ExcitationSolve features: First, using ExcitationSolve to select new operators leads to fewer operators being added to the ansatz. This results in a shallower circuit and a cheaper re-optimization. This can be seen for the larger molecules, i.e., LiH in Fig. 7c, where ADAPT-VQE requires 34 operators, while ExcitationSolve only needs 30 to converge. For $H_2O$, ADAPT-VQE needs 48 operators, while ExcitationSolve only requires 42. Note that the operator reduction mostly but not only becomes significant beyond chemical accuracy. Second, initializing the new operators with ExcitationSolve at their optimal values offers a beneficial warm start for the intermediate parameter optimization, further leading to a convergence within fewer iterations in the re-optimization of the parameters. A special

case can be seen for $H_2$ in Fig. 7a, where only a single excitation contributes to the ground state and ExcitationSolve immediately initializes it with its optimal parameter value. The most significant reduction in energy evaluations can be observed for $H_2O$ where ExcitationSolve reaches the chemical accuracy approximately 15 times faster. Supplementary Note 2.2 adds resource comparisons. One might wonder whether the reduction in selected operators is due to the ExcitationSolve parameter optimizer or the ExcitationSolve operator selection – or perhaps even a joint effort. An in-depth analysis in Supplementary Note 2.3 shows that the reduction stems from the ExcitationSolve selection. The fastest convergence is still achieved by employing ExcitationSolve for both tasks.

**Dissociation curves**. We further analyze how a deviation of the HF state from the actual ground state influences the performance of ExcitationSolve compared to GD, BFGS and COBYLA when provided as the initial state in the fixed UCCSD ansatz VQE. Concretely, by varying the inter-atomic distances in the molecules, we affect how closely the HF state approximates the true ground state: The further the bond is stretched, the larger the initial HF error $|E_{HF} - E_{FCI}|$ of the HF energy $E_{HF}$ to the energy of the FCI solution $E_{FCI}$, signaling the emergence of *strong correlations*[50]. This HF error dependence on the bond distance for all studied molecules is shown in Fig. 8.

Figure 8 shows how many energy evaluations are needed for different bond distances to reach convergence for each molecule. See Supplementary Note 2.4 about the final energy differences to FCI when convergence is reached. Among all molecules it becomes apparent that the higher the bond distance gets, i.e., the higher the HF error since the initial HF state deviates more from the ground state solution, the more executions of the circuit are necessary to find the ground state. We see only two exceptions: for $H_2$ (Fig. 8a) the ExcitationSolve optimizer finds the ground state with a constant number of evaluations and for $H_3^+$ (Fig. 8b) ExcitationSolve also needs a constant number of evaluations when optimizing two parameters at the same time. These two cases are special, because (2D-)ExcitationSolve can set the parameters to the global optimum within the first VQE iteration because the ground state of $H_2$ and $H_3^+$ are superpositions of the HF state with one and two excited states, respectively. Overall, it can be observed that ExcitationSolve outperforms the other optimizers for each bond distance. Although the relative difference in the number of energy evaluations needed for the optimizers to reach convergence is mostly independent of the bond distance. Indeed, the scaling with the bond length for the two larger molecules LiH (Fig. 8c) and $H_2O$ (Fig. 8d) is almost identical for all optimizers, with BFGS seeming least impacted by the bond distance. Finally, see Supplementary Note 2.5 for a closer look at certain large $H_2O$ bond lengths, where ExcitationSolve faces convergence issues in local minima and flat regions, and how strategies like ExcitationSolve2D and parameter shuffling help overcome these challenges.

**NISQ hardware benchmarks**. To conclude the experimental evaluation of ExcitationSolve, we examine the near-term applicability of ExcitationSolve by repeating previous experiments on the IBM quantum computer `ibm_quebec`, which is referred to as IBM-Q henceforth. This quantum processor features the *Eagle r3* architecture with 127 qubits and classifies as a noisy intermediate-scale quantum (NISQ) device. Decoherence over time significantly impedes the execution of quantum circuits of increased depth, primarily limited by the two-qubit gate count. Therefore, the implementation of excitation operators on current NISQ devices is a challenge due to their decomposition into rather deep circuits over a hardware-native gate set[26]. This is a limitation of these ansätze rather than ExcitationSolve or any other optimizer per se.

Nevertheless, we select $H_2$ and $H_3^+$ for fixed UCCSD ansatz IBM-Q experiments, analogous to the "Fixed ansatz (UCCSD) comparison with other optimizers" subsection. The results are depicted in Fig. 9 and discussed in the following. For a comparison, simulated experiments with pure shot noise are presented in Supplementary Note 2.6. In Supplementary Note 2.7, LiH serves to study adaptive ansätze by focusing on the robustness of the

 

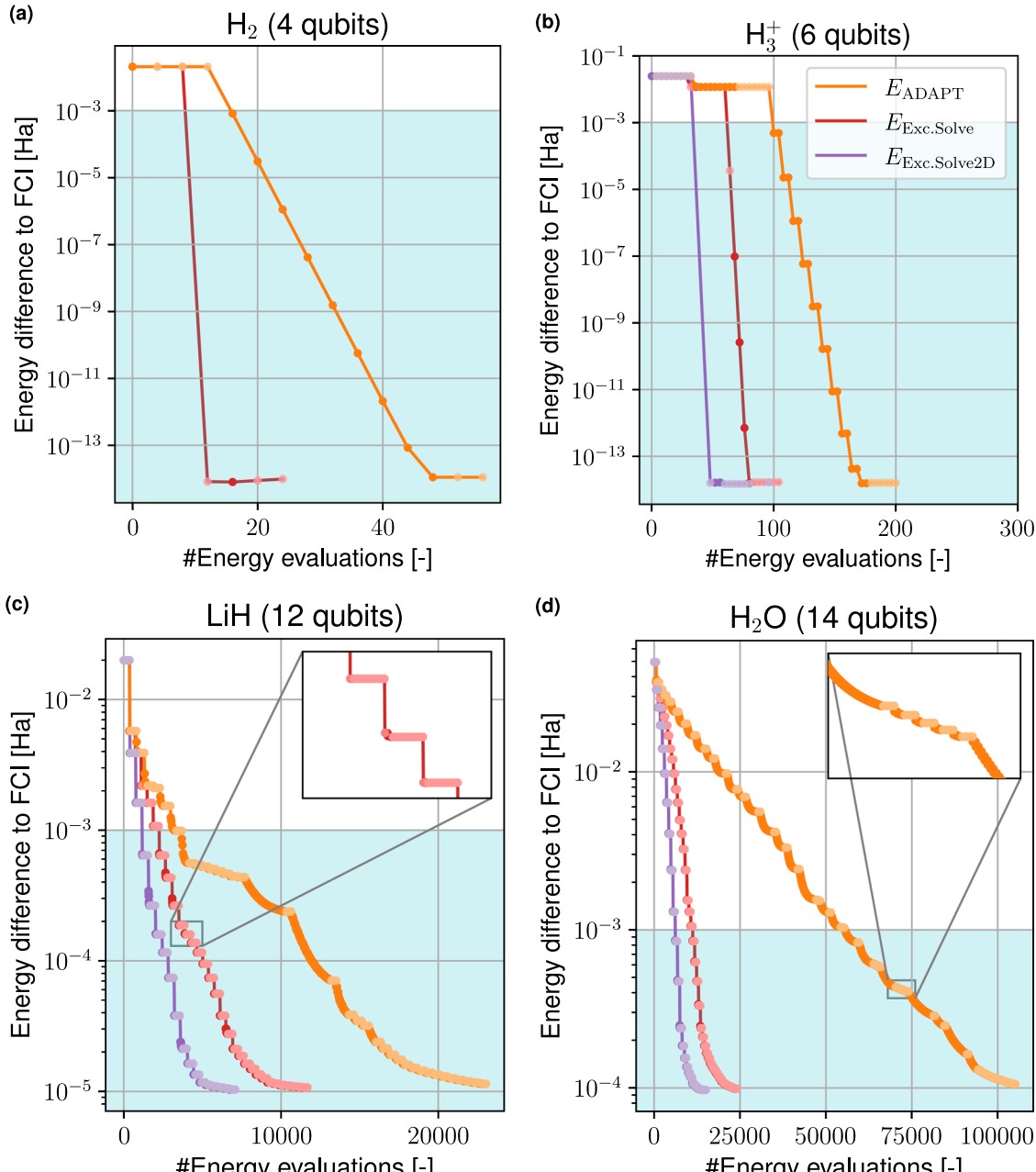

**Fig. 7 | Comparison between ExcitationSolve and original ADAPT-VQE with the UCCSD operator pool.** The plots show the error of the VQE with respect to the Full Configuration Interaction (FCI) solution $|E_{VQE} - E_{FCI}|$ over the number of energy evaluations for the molecules **a** $H_2$, **b** $H_3^+$, **c** LiH and **d** $H_2O$. Lighter plot colors signal evaluations needed for operator selection, darker plot colors mark optimization steps, with insets magnifying regions of interest. The light blue region signifies the chemical accuracy ($10^{-3}$ Ha).

operator selection criterion of ExcitationSolve compared to the original ADAPT-VQE on a NISQ device.

Over the five repeated experiments for both, $H_2$ (Fig. 9a) and $H_3^+$ (Fig. 9b), ExcitationSolve demonstrates strong robustness against hardware noise by reproducing the convergence within a single iteration as observed in both exact and shot noise simulation. Convergence means that the parameters that were found by the optimizers by solely using energy (or gradient) information from the IBM-Q prepare the ground state within chemical accuracy when re-evaluating the associated energy *via exact simulation*. ExcitationSolve hereby surpasses GD in both speed and quality: in all five repetitions of the $H_2$ experiment, ExcitationSolve reaches chemical accuracy within the first iteration, even within the 95% confidence intervals. The GD step size was tuned in preliminary $H_2$ IBM-Q experiments and chosen as the largest non-diverging step size tested.

Even for the more challenging $H_3^+$, ExcitationSolve still achieves chemical accuracy within the first iteration on average, which was no longer observed for GD in the given time. COBYLA was not examined for $H_3^+$ after it proved incapable of handling the hardware noise in the smaller $H_2$ case, where no improvement over the HF energy was achieved. COBYLA does not preserve sparsity in the iterates, i.e., does not keep most parameters zero, which leads to deeper and thus noisier transpiled circuits. This is in contrast to ExcitationSolve and GD, which owe some of their success under hardware noise to this sparsity, which allows smaller quantum circuits to be executed on the IBM-Q. As a conclusion, the successful convergence, especially via ExcitationSolve within a single iteration, is striking and rather unexpected given the relatively high errors in the energy estimates obtained from the IBM-Q device, which the energy reconstructions were derived from. To put this into perspective,

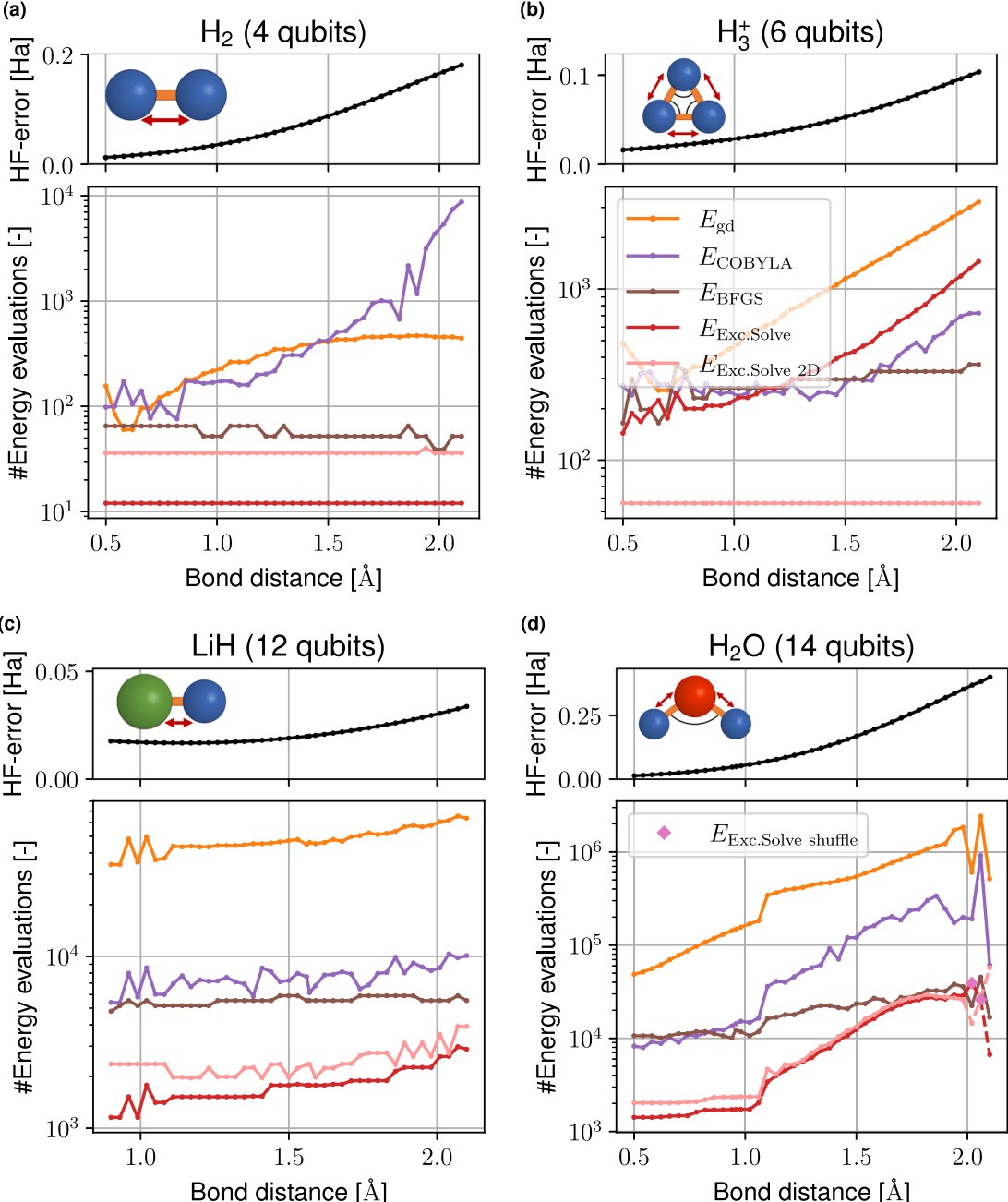

**Fig. 8 | Comparison of optimizers using fixed UCCSD ansätze for non-equilibrium geometries.** Hartree-Fock (HF) error (black) and energy evaluations until VQE convergence in dependence of the bond length for optimizers ExcitationSolve (red), COBYLA (purple) and Gradient descent (yellow) for molecules **a** $H_2$, **b** $H_3^+$, **c** LiH and **d** $H_2O$. For $H_2O$ with high bond lengths, random shuffling of the parameter order in ExcitationSolve is additionally utilized to achieve convergence (diamond marker). The HF error is the absolute difference between the Full Configuration Interaction (FCI) energy and HF energy.

not only is the deviation typically outside the chemical accuracy, but the absolute error may even exceed the initial HF error. Nevertheless, the parameters that ExcitationSolve optimized based on these noisy inputs prove to prepare the ground states within chemical accuracy. A reason why the reconstruction remains useful is that the hardware noise introduces a systematic error. This error may manifest as a rescaled amplitude or an offset shift of the reconstructed finite Fourier series, as depicted for $H_3^+$ in the inset plot of Fig. 9b. Importantly, these transformations do not affect the location of the minimum, leading to parameter values that still prepare the ground state accurately. Estimating the energy for the final ExcitationSolve parameters on IBM-Q with a shot budget exceeding 8192, which was otherwise used for energy evaluations, reveals that an at

least five-fold shot increase achieves chemical accuracy for $H_2$, whereas no such improvement – not even below the HF error – is observed for the more complex $H_3^+$. Beyond $H_3^+$, we observed hardware noise to obscure energy information due to an increased circuit depth from multiple excitations being active in the ansatz. Therefore, the error is not systematic enough to allow for sufficient energy reconstructions.

## Discussion
The main motivation behind ExcitationSolve is to unite the benefits of quantum-aware optimization and physically-motivated ansätze composed of excitation operators. Due to the quantum-awareness in particular, ExcitationSolve improves over common gradient-based optimizers by

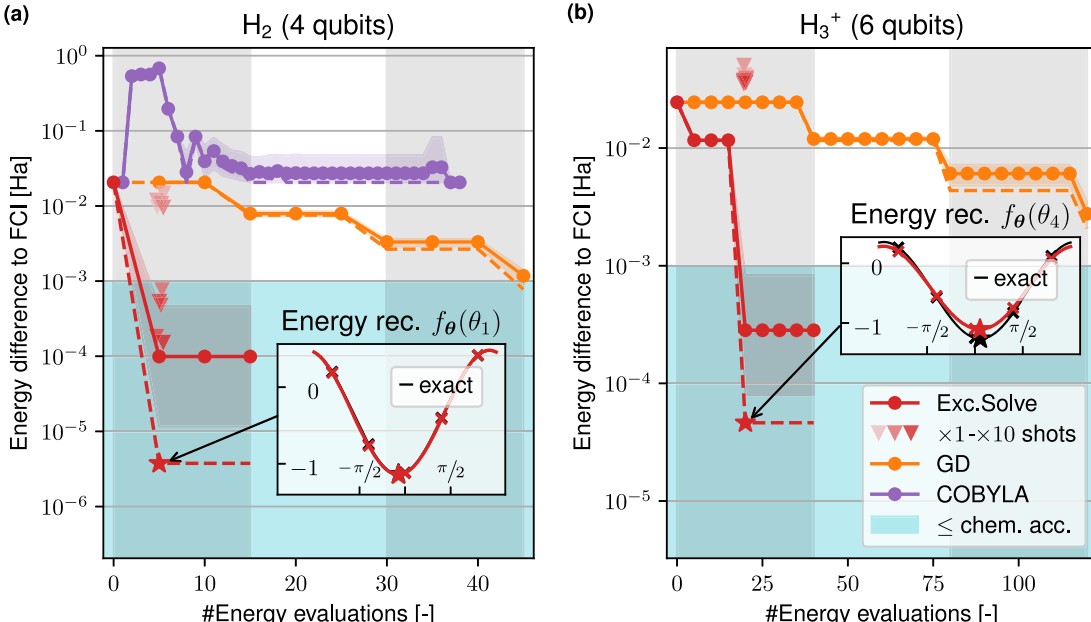

**Fig. 9 | Benchmarks on NISQ (`ibm_quebec`) quantum processor for fixed UCCSD ansätze.** The molecules **a** $H_2$ and **b** $H_3^+$ are studied, analogous to the simulated experiments in Fig. 6a, b, respectively. The optimizers considered are ExcitationSolve (red), Gradient descent (yellow), and COBYLA (purple), where the latter was discarded from the $H_3^+$ experiments. Per optimizer, five experiments are performed out of which the best run (dashed), along with the mean (solid) and 95% confidence intervals (bands), are presented in terms of the VQE energy with respect to the Full Configuration Interaction (FCI) solution $|E_{VQE} - E_{FCI}|$. While all presented optimizations exclusively rely on energy value (and gradient) information extracted from the the NISQ device `ibm_quebec`, the FCI errors shown are based on exact re-evaluations via state vector simulation for a clear quality assessment. Triangles instead represent such `ibm_quebec` energy estimates in the final ExcitationSolve parameter configuration (star), with shot counts from 1× (light red) to 10× (dark red) of the 8192 default. The inset plots also show the noisy energy values (red crosses) and compare the resulting energy reconstructions $f_\theta(\cdot)$ (red) as used by ExcitationSolve with the exact energy function (black). The light blue background signifies chemical accuracy. Vertical lines mark the iterations in ExcitationSolve and GD.

**Table 1 | Number of shifts comparison for the full single-parameter energy landscape reconstruction vs. the partial derivative via a parameter-shift rule**

| | Function reconstruction | | Partial derivative via parameter-shift rule | |
| --- | --- | --- | --- | --- |
| | *Theoretical* | *Effective* | *Complex WF* | *Real WF* |
| Rotations ($G^2 = I$) | 3[22,23] | 2[22,23] | 2[60,61] | – |
| Excitations ($G^3 = G$) | 5 | 4 | 4[62] | 2*[42] |

Reconstruction numbers distinguish the *theoretical* minimum and the *effective* number after reusing previously evaluated energies. Note that the number of shifts for excitations on real-valued wavefunctions (*) does not exactly correspond to pure energy evaluations[42].

informing updates globally instead of being limited to the local vicinity—without imposing any resource overhead and, on top of that, circumventing any hyperparameter tuning. Specifically, reconstructing the energy landscape for a single parameter requires four energy evaluations, optimized by reusing the final minimal energy from the previous step, which matches the resource requirements for the four-term parameter-shift rule. Interestingly, the same resource-efficiency already can be observed with simpler rotations in Rotosolve. When considering real wave functions (real-valued quantum states), as we did for all examples in the Results, a more efficient approach which manages with only two shifts exists[42] This approach is, however, at the cost of circuit modifications as detailed in Supplementary Note 5.1. Table 1 provides a resource comparison.

In the adaptive setting, we globally inform the operator selection criterion with ExcitationSolve by using the same quantum resources more effectively compared to the original gradient criterion in ADAPT-VQE[21]. We remark that any double excitation can be decomposed into a product of 8 Pauli rotations of the same angle[26]. A naive application of SMO/Rotosolve to the Pauli decomposition leads to an overestimation of the Fourier order by a factor of 4. This further worsens for higher-order excitations where SMO exponentially overestimates the Fourier order. However, the more sophisticated decomposition of excitation operators into *two* commuting self-inverse operators (cf. Supplementary Note 4 and ref. 42) reveals that ExcitationSolve can be interpreted as the double-occurrence case of SMO.

Moreover, the relevance of ExcitationSolve is intrinsically linked to the advantages of employing physical ansätze using excitation operators. The significance of such ansätze lies in their ability to preserve essential physical quantities and symmetries. It could be argued that *qubit tapering*[51,52] for hardware-efficient or problem-agnostic ansätze, i.e., composed of rotation gates, could achieve the same advantages; however, qubit tapering conserves the particle numbers and spin symmetries only up to their parity. Note that ExcitationSolve natively handles excitation operators, independent of the actual decompositions, fermion-to-qubit mappings and simplifications of the ansatz to the quantum circuit. In addition, organizing the parameters in a problem-informed way via such physical excitation operators could lead to a simpler, more suggestive optimization landscape as it has been observed in other contexts[53]. Alternatively, the choice of physically-motivated ansätze can be interpreted as encoding an *inductive bias*, and, hence, could effectively restrict the exponentially growing underlying Hilbert space. This potentially counteracts the emergence of so-called *barren plateaus*, which would obstruct the practical use in realistic, large-scale problem sizes[49].

Our experimental results demonstrate multiple advantages of ExcitationSolve over previous state-of-the-art optimizers, including gradient descent (GD), COBYLA[18], Adam[13], SPSA[19,20], and BFGS[14–17]: Firstly, for a fixed UCCSD ansatz, ExcitationSolve generally takes fewer iterations to converge to the ground state than any other tested optimizer. How significant the advantage is depends on the molecule. Secondly, ExcitationSolve has no hyperparameters that need to be tuned and needs no calibration. Thirdly, in an adaptive setting, like ADAPT-VQE[21], ExcitatonSolve can be used to choose the next operator to append based on its highest impact on the energy value outperforming the locally informed selection such as the original ADAPT-VQE gradient criterion. The newly picked operator is also already initialized in its optimal parameter value, which significantly warm-starts the intermediate optimization of all present parameters before extending the ansatz further. In NISQ experiments on the IBM quantum device, ExcitationSolve shows robustness against both shot- and hardware noise, and preserves the advantages over other optimizers that were observed in simulation. Decoherence errors ultimately impose a limit but are natural for the depth of transpiled circuits from ansätze composed of excitation operators, which is independent of the chosen optimizer.

While ExcitationSolve offers promising features, it faces certain challenges, such as the potential for getting trapped in a local minimum. This issue becomes particularly significant for larger molecules, where the proliferation of local minima poses a critical challenge. To address this, we proposed a multi-dimensional extension of ExcitationSolve, though this approach introduces new complexities, such as increased computational cost, as the number of evaluations grows exponentially with the number of parameters. Moreover, determining which parameters to pair for optimization is complex due to the combinatorial nature of the problem. One effective heuristic is to pair operators with the strongest impact from single-parameter optimization, as demonstrated with the $H_3^+$ molecule. In the case of $H_2O$, we observe that this heuristic is not always sufficient to avoid local minima. Fortunately, we find that parameter shuffling provides another, complementary approach to avoid local minima – and unlike the 2D optimizer does not even require additional computational effort. We did not find a systematic way of deciding when to apply shuffling to avoid local minima. But, what we find is that the order in which the parameters in the UCCSD ansatz are optimized can, at least for ExcitationSolve, play a crucial role in overall convergence. One could investigate if turning parameter shuffling on and off during optimization can increase convergence speed or even avoid local minima. Furthermore, one needs to analyze what causes the 1D optimization to settle in a local minimum and why parameter shuffling is able to circumvent it. In general, ExcitationSolve cannot be guaranteed to find a global minimum. Hence, identifying strong heuristics for beneficial parameter orderings and for how and when multi-parameter optimization can be employed to navigate the energy landscape faster and more reliably remains an area for future investigation.

Besides the showcased application of ExcitationSolve to ground state preparations, it can be analogously employed to perform projected variational quantum dynamics (pVQD)[54]. The utility of gradient-free optimization for time evolution has already been demonstrated[55] via Rotosolve for hardware-efficient ansätze to replicate a Trotter step. The applicability of ExcitationSolve to excitation-based ansätze for pVQD becomes apparent when reformulating the overlap maximization with the zero-state $|0\rangle$ as the minimization of the Hermitian zero-state projector $P = |0\rangle\langle 0|$[56], taking the role of the Hamiltonian $H$ in this work.

## Methods
### ExcitationSolve for multiple occurrences of parameters
For the practical use of UCCSD, it often suffices to employ a single time step in first-order Trotterization. However, one might encounter scenarios where the expressivity of such type of ansatz is no longer sufficient and thus requires a refinement—e.g., through a higher order product formula or simply multiple time steps. In a higher order product formula, at least one or more parameters appear multiple times

throughout the corresponding quantum circuit. Concerning the use of multiple time steps, there is the degree of freedom of using different parameters for every time step or sharing these parameters between multiple steps. The latter is motivated by the physical point of view of a discretized time evolution of an adiabatic process with equal time steps where the strength of the free fermionic problem, i.e. the single-excitations, is kept constant. For $S$ excitations sharing the same variational parameter $\theta$,

$$f_\theta(\theta) = \sum_{s=1}^{2S} a_s \cos(s\theta) + \sum_{s=1}^{2S} b_s \sin(s\theta) + c, \quad (9)$$

i.e., it is a one-dimensional Fourier series of order $2S$. The $4S+1$ coefficients can be determined through $4S$ energy evaluations. This result can be straightforwardly inferred from the multi-parameter generalization from Eq. (8) by setting multiple parameters equal and reducing the trigonometric form. Equation (9) is closely related to the result for multiple occurrences of a single parameter in Rotosolve/SMO, where the energy landscape is given by a Fourier series of order $S$ with $2S+1$ coefficients[23] (cf. Supplementary Note 5.2). For a detailed derivation, refer to Supplementary Note 4.4. Note that unlike the exponential growth in energy evaluations for a multi-parameter optimization, we have a linear growth for the multi-occurrence case. Supplementary Note 3.3 presents the most generic energy landscape form when considering multiple such repeated parameters simultaneously.

### Classical minimization of analytic energy reconstructions
After reconstructing the energy function, in order to determine the global minimum in the single parameter (or set of parameters) classically, we suggest the utilization of the following approaches.

**Single-parameter case (companion-matrix method).** The first important realization is that the derivative of the finite Fourier series determining the energy function in one parameter as in Eq. (3), is again a Fourier series of the same order, i.e., $\frac{d}{d\theta} b_s \sin(s\theta) = b_s s \cos(s\theta)$, $\frac{d}{d\theta} a_s \cos(s\theta) = -a_s s \sin(s\theta)$ and zero constant. Of this derivative function we can determine the zeros and evaluate the analytic energy function (classically) at these points. The smallest of the resulting energy values must be the global minimum.

Finding the zeros of the derivative can be achieved efficiently through the so-called companion-matrix method. In the following we provide a brief review of this technique introduced in Ref. 57 for a general (finite) Fourier series. Note that the constant term $c = 0$ as we deal with minima, i.e., zeros of derivatives:

$$f(\theta) = \sum_{s=1}^{S} a_s \cos(s\theta) + \sum_{s=1}^{S} b_s \sin(s\theta). \quad (10)$$

By employing the Euler identity, this finite Fourier series may be recast into the complex form

$$
\begin{aligned}
f(\theta) &= \sum_{s=1}^{S} \left( \frac{a_s - ib_s}{2} e^{is\theta} + \frac{a_s + ib_s}{2} e^{-is\theta} \right) \\
&= e^{-iS\theta} \sum_{s=1}^{S} \left( \frac{a_s - ib_s}{2} e^{i(S+s)\theta} + \frac{a_s + ib_s}{2} e^{i(S-s)\theta} \right) \\
&= e^{-iS\theta} \left( \sum_{s=S+1}^{2S} \frac{a_{s-S} - ib_{s-S}}{2} e^{is\theta} + \sum_{s=0}^{S-1} \frac{a_{S-s} + ib_{S-s}}{2} e^{is\theta} \right).
\end{aligned}
\quad (11)
$$

By introducing the transformation $z = e^{i\theta}$, we rewrite the Fourier series as

$$f(\theta) = \frac{z^{-S}}{2} \sum_{s=0}^{2S} h_s z^s =: \frac{z^{-S}}{2} h(z), \quad (12)$$

where

$$h_s = \begin{cases} a_{S-s} + ib_{S-s}, & s = 0, 1, \ldots, S-1 \\ 0, & s = S \\ a_{s-S} - ib_{s-S}, & s = S, S+1, \ldots, 2S. \end{cases} \quad (13)$$

and $h(z)$ is referred to as the *associated polynomial*. (Since $\bar{h}_s = h_{2S-s}$ holds, $h(z)$ is a complex *self-reciprocal* or *palindromic* polynomial.) Note that the problem of finding the real zeros of $f(\theta)$ has now been transformed to the task of determining the zeros of $h(z)$ on the complex unit circle. To solve for the roots of the associated polynomial, the $2S \times 2S$ companion matrix/ Frobenius matrix $B$ is constructed with the component in the $s$-th row and $t$-th column

$$B_{st} = \begin{cases} \delta_{s,t-1}, & s = 1, 2, \ldots, 2S-1 \\ -\dfrac{h_{t-1}}{a_S - ib_S}, & s = 2S, \end{cases} \quad (14)$$

where $\delta$ denotes the Kronecker delta. The characteristic polynomial of the companion matrix is precisely the associated polynomial from before. In the case of ExcitationSolve, where $S = 2$, the companion matrix takes the form

$$B = \begin{pmatrix} 0 & 1 & 0 & 0 \\ 0 & 0 & 1 & 0 \\ 0 & 0 & 0 & 1 \\ -\dfrac{a_2 + ib_2}{a_2 - ib_2} & -\dfrac{a_1 + ib_1}{a_2 - ib_2} & 0 & -\dfrac{a_1 - ib_1}{a_2 - ib_2} \end{pmatrix} \quad (15)$$

The roots $\theta_k$ of $f$ are then obtained as

$$\theta_k = \arg(z_k) + 2\pi m - i \log(|z_k|), \quad (16)$$

where $z_k$ is the $k$-th eigenvalue of $B$ and $m$ is some integer. Here, it becomes clear that $\theta_k$ is real iff $z_k$ lies on the complex unit circle.

**Multi-parameter case (Nyquist initialization and local optimization).** In the case of an analytic energy function in multiple parameters to be optimized as in Eq. (8), we can no longer employ the companion-matrix method.

One naive way to find the minimum of a multiple-dimensional energy landscape lies in rasterizing of the parameter space. For low dimensions, such a brute-force evaluation can be easily performed on a classical computer (keep in mind that the energy function has already been faithfully reconstructed and can be evaluated at arbitrary positions in parallel). For this type of approach, however, the precision of the result is limited by the resolution of the grid, which is not desirable as the precision then depends precisely on the position of the minimum and cannot be assumed to be constant.

Inspired by the Nyquist-Shannon sampling theorem[58], we conjecture that it is sufficient to evaluate the energy function with a lattice spacing of $\Delta = 2\pi/(2\omega_{\max} + 1)$, where $\omega_{\max} = 2S$ is the highest frequency in the system along an $S$-fold occurring parameter, i.e., we sample with at least twice the highest frequency of the system (Nyquist frequency), which is $2\omega_{\max} + 1$ equidistant samples within the period. Each of those lattice points is then taken as an initial guess for a local optimization scheme such as gradient descent. This can be implemented on classical hardware very efficiently for the following reasons. Firstly, all runs for the different initial guesses can be performed in parallel. Secondly, since the function to be minimized is known analytically, analytical gradients are also readily available. Thirdly, through an optimal gradient descent step size $\alpha$, the convergence is guaranteed and the convergence speed can be improved significantly with $\alpha = 1/L$ where $L$ denotes a Lipschitz constant of the gradient[59], which can be determined given the analytic form as a multi-

dimensional Fourier series. Bear in mind that, while this technique performs much more efficiently and precisely than a naive high-resolution grid evaluation, its computational costs still scale exponentially in the number of (different) parameters.

## Data availability

Data sets generated and analyzed during the current study are available from the corresponding author upon reasonable request.

## Code availability
The source code of the ExcitationSolve algorithm is available on GitHub: https://github.com/dlr-wf/ExcitationSolve. This includes the standard algorithm for fixed and adaptive ansätze, as well as the two-dimensional variant. Some minimal examples are provided to make the experiments conducted in this work reproducible.

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

## Acknowledgements

We thank Jakob S. Kottmann and David Wierichs for fruitful discussions. We would also like to thank Thomas Plehn and Gabriel Breuil for helpful comments on the manuscript. Furthermore, we would like to extend our gratitude to the anonymous referees for their helpful suggestions throughout the peer review process to enhance the presentation and evaluation of our work. This project was made possible by the DLR Quantum Computing Initiative and the Federal Ministry for Economic Affairs and Climate Action; https://qci.dlr.de/quanticom (J.J., T.N.K., M.H., and E.S.). We further acknowledge the NSERC CREATE in Quantum Computing Program, grant number 543245 (J.J.). Finally, we acknowledge the use of IBM Quantum services for this work via the IBM Quantum System One Access Program funded by the Quantum Algorithms Institute (QAI) (J.J.). The views expressed are those of the authors, and do not reflect the official policy or position of IBM or the IBM Quantum team.

## Author contributions

J.J. initiated and managed the project and devised the main concepts and initial proofs. T.N.K. worked out the large majority of the theoretical proofs. All authors conceived and planned the experiments. J.J. implemented the methods, while M.H. and E.S. wrote the simulation code. M.H. and E.S. carried out the statevector simulation experiments. J.J. and T.N.K. carried out the quantum hardware experiments. All authors analyzed the data and contributed to the interpretation of the results. All authors contributed to writing the manuscript, with J.J. and T.N.K. leading the effort.

## Funding

## Competing interests

A patent application filed by the German Aerospace Center (Deutsches Zentrum für Luft- und Raumfahrt e.V., DLR), currently pending with the German Patent and Trade Mark Office (Deutsches Patent- und Markenamt, DPMA), covers aspects of this work. It specifically includes, but is not limited to, the ExcitationSolve method for fermionic excitations and fixed ansätze. The listed inventors are identical to the authors of this work. The application number is DE 10 2024 115 387.3, with the German title "Verfahren zur Bestimmung von Energien und Energiezuständen". The authors declare no other financial or non-financial competing interests.
