## [Transparent Peer Review file · Communications Physics]

Fast gradient-free optimization of excitations in variational quantum eigensolvers

Corresponding Author: Mr Jonas Jäger

Version 0:

Reviewer comments:

Reviewer #1

(Remarks to the Author)

This manuscript proposes a gradient-free quantum optimization method tailored to ansätze constructed from excitation operators. The authors name this method ExcitationSolve, and it can be viewed as an extension of the Greedy Gradient-Free algorithm developed by Piquemal et al. While Piquemal et al. applied their method to Pauli operators appearing in UCCSD, the present study applies the optimization directly to the excitation operators themselves. At first glance, the idea may appear incremental, but its implementation is well motivated within the framework of quantum chemistry, and the method demonstrates significantly improved performance compared to conventional approaches. This advantage is observed not only in ideal simulations but also in the presence of shot noise and on real quantum devices. In light of these promising results, the manuscript has potential for publication, provided that several concerns are addressed.

1. The authors present their approach as a new method; however, it is essentially a variant of the optimization techniques developed in Rotosolve and by Piquemal et al. It would be preferable to adopt a consistent naming scheme that clearly reflects its connection to the method of Piquemal et al., so that readers can readily recognize it as an extension rather than an entirely new approach.

2. The authors claim that “the average estimate variance becomes inversely proportional to the total shot budget and, importantly, independent of the number of evaluation points.” However, prior work by Endo et al. demonstrates that other gradient-free methods—such as Free-Axis Selection (Fraxis), Free Quaternion Selection (FQS), and optimal parameter combinations—can achieve maximal optimization for a single gate without relying on gradients. Based on these findings, Endo et al.[58] conclude that allocating more shots to a single parameter configuration is preferable. The authors clearly should discuss this inconsistency.

3. The authors state that “...this approach can make ExcitationSolve more robust against noise...” and that “...least-squares estimation yields the optimal result under a normally distributed noise assumption... which is approximately fulfilled for pure states.” However, while the proposed optimizations may be locally optimal, this does not necessarily guarantee convergence to a global minimum. To convincingly demonstrate the practical utility of these gradient-free local optimization methods, it is important to accumulate and present a range of empirical evidence. In fact, similar observations have already been reported in related methods such as Rotosolve/NFT, Free-Axis Selection and Free Quaternion Selection. We recommend that the authors emphasize the practical relevance of their approach by referring to these prior works.

4. In Fig. 6, which shows noiseless simulations using a fixed-depth UCCSD ansatz, the energy sometimes increases even though analytical minimization is used in each ADAPT step. If the optimizer is truly analytical, the energy should not increase under ideal conditions. This behavior seems inconsistent and should be explained.

5. In addition, the authors discuss the method’s robustness to shot noise; however, this does not necessarily extend to noise on real quantum devices. In the real-device experiment shown in Figure 9, the authors report that the energy estimated directly from the circuit is higher than that of the Hartree–Fock state, yet the optimal parameters themselves can still be identified. This appears to be intuitively contradictory. If this observation is accurate, does it imply that real-device noise introduces systematic errors? If not, would this favorable behavior persist as the system size increases? The authors should address this issue in more detail.

Reviewer #2

(Remarks to the Author)

The authors propose a new optimization method, termed ExcitationSolve, aimed at optimizing ansätze for the electronic structure problem. In particular, the optimizer uses the structure of fermionic (or qubit) excitations to exactly minimize one-dimensional cost landscapes based on just a few energy evaluations per parameter. This generalizes the well-known RotoSolve algorithm, which applies to ansätze composed of unitaries generated by individual Pauli strings. The authors compare their protocol against optimization methods in the literature and find that it significantly reduces the number of function evaluations required to reach a given accuracy. Based on hardware experiments on up to 6 qubits, the protocol is shown to also be more resilient to noise. I believe that the work is relevant and could recommend publication once the remarks below are addressed.

- Following the introduction of ADAPT-VQE, the authors claim that "the VQE optimization problem is generally challenging" and that there is a "large number of local minima" which leads optimizers to fail. However, this is the opposite of what we observe in ADAPT-VQE landscapes - the parameter initialization strategy seems to effortlessly lead to good quality minima that are easily found by local optimizers. See <https://www.nature.com/articles/s41534-023-00681-0>

- When the authors state that each parameter must occur only once in the ansatz for Eq. 3 to hold, they should specify that they mean that the excitation must only occur once in the ansatz with this parameter. Decomposing an excitation into basic gates will in general require more than one single-qubit rotation gate with this rotation angle, but I believe this will not hinder the application of the formula, as we can see the implementation of the unitary as a block in the circuit, regardless of the actual physical implementation.

- The description of the algorithm in the main text and in the scheme of Fig. 2 is a bit confusing. The authors use k to label the iteration and j to label the operator whose coefficient is modified in this iteration, but they do not specify which j is optimized in each iteration. From the supplementary material, it seems that parameters are updated in order, with each individual update counting as one iteration, and the optimization continues cyclically until a certain convergence criterion is met. This should be clear from the main text and the flowchart in Fig. 2.

- The variable N that appears in the convergence criterion does not seem to be defined.

- The authors focus the discussion on page 6 on occupied-to-virtual excitations, but it's worth mentioning generalized excitations since the algorithm readily applies to those also.

- The use of the expression "globalized ADAPT-VQE operator selection criterion" might confuse the readers, since the word "globalized" might suggest that this criterion takes into account the interplay between the parameters in a global optimization, which is not the case. Similarly, the sentence "We select the operator that achieves the strongest decrease in energy to be appended to the ansatz" is misleading, since the strongest decrease in energy mentioned here is the one achievable by changing only one variational parameter. The strongest energy decrease that might be produced by adding an operator and performing a full optimization remains unknown. I would suggest making this clear. While this is discussed later, even just rephrasing this as "that achieves the strongest decrease in energy when appended to the previous ansatz with a fixed coefficient" might avoid misconceptions.

- In Fig. 4, it is unclear whether the example provided corresponds to an actual scenario observed in numerical simulations or if it is fabricated to showcase the motivation - could the authors make this clear?

- I believe that the selection criterion deserves more attention. The authors claim that their method is superior, but it is difficult to assess this when the selection criterion is implemented along with the optimization method. Comparing ADAPT-VQE with the gradient vs energy-based selection criterion using the same optimizer would be ideal.

- In Fig. 5, the authors should include the scale of energy/error values associated with the colors. The final 1D and 2D errors should also be included to show the magnitude of the benefit of using the 2D optimization.

- The authors should specify the order of the excitations they consider for the UCCSD ansatz.

- More details should be provided about the basis set considered for the molecules. It would also be helpful to mention the number of qubits used to represent each in the main text in addition to the plots.

- Systems are restricted to be small and/or weakly correlated, in particular because bond distances are set to equilibrium. It would be interesting to test stretched bond distances and slightly larger, strongly correlated molecules such as H₆. I appreciate that the authors include bond dissociation curve plots for the UCCSD ansatz, but (1) they do not consider the same for ADAPT-VQE, (2) they do not include error plots, and (3) they do not consider a staple difficult system such as H₆. In particular, H₆ at stretched bond distances is known to lead to a roadblock in ADAPT-VQE known as gradient troughs (see <https://www.nature.com/articles/s41534-023-00681-0>). It would be interesting to understand whether ExcitationSolve could help mitigate this problem.

- The authors plot the energy error against the number of energy evaluations, but it is not clear how they factor the gradient evaluations into the costs for the case of gradient-based optimizers.
- In Fig. 6, it is unclear why methods such as BFGS and GD follow this pattern, where they are constant across one vertical shading. According to the text, the shading marks one iteration over all parameters - what does this mean? One iteration will require a different number of function evaluations depending on the method, so it is strange that several optimizers have a constant energy value across these shadings.
- When comparing the number of function evaluations required across the bond dissociation curve in Fig. 8, it would be relevant to include the final VQE error in addition to the HF error.
- In Table I, I suppose that the "theoretical" vs "effective" number of energy evaluations required to reconstruct the function comes from the fact that the energy value for the unshifted coefficient can be recycled from previous calculations, but it might be worth clarifying and referring to earlier statements concerning the subject.
- In table I, the authors should note that while general parameter-shift rules for these generators require 4 circuit evaluations per parameter, this might be reduced to 2 if the wavefunction is real, which is the case here. Further, writing the cost of the parameter shift rules in terms of the number of energy evaluations is not accurate, because the excitation generators have three eigenvalues. This difference in the spectrum leads the PSR formulas to require circuits that differ from the ones necessary to evaluate the energy. This is discussed in Ref. 35.
- The authors refer to their experiments as being performed "on a 127-qubit IBM quantum device", which is misleading considering that the experiments only use up to 6 qubits. The full size of the device is not relevant for the experiments performed.
- Parameter shuffling seems necessary to achieve convergence in the largest molecule, but I am concerned that there is no systematic way of deciding when to apply this. The manuscript does not seem to include enough details about when to apply shuffling or what exactly this consists of.
- Most of the Methods section ("ExcitationSolve for multiple occurrences of parameters", "Classical minimization of analytic energy reconstructions", " Reconstruction strategies for noise robustness") could be moved into the supplementary material, since it mostly consists of technical details which are peripheral to the results and the manuscript could benefit from being more succinct.

Reviewer #3

(Remarks to the Author)

Please find the attached comments

Reviewer #4

(Remarks to the Author)

I co-reviewed this manuscript with one of the reviewers who provided the listed reports. This is part of the Communications Physics initiative to facilitate training in peer review and to provide appropriate recognition for Early Career Researchers who co-review manuscripts.

Version 1:

Reviewer comments:

Reviewer #1

(Remarks to the Author)

The authors have adequately addressed the concerns raised in the previous round of review.

With these revisions, the paper presents a well-motivated and practically useful method for variational parameter optimization in quantum algorithms. The theoretical framework is sound, the implementation is efficient, and the results are compelling both in simulation and on real hardware.

I therefore recommend acceptance of this manuscript for publication.

Reviewer #2

(Remarks to the Author)

I appreciate the improvements to the manuscript, which addressed many of the questions I had. Once the two points below have been addressed, I can recommend publication.

- While I agree with the authors' statement that "extensive analyses provides convincing evidence for our conclusions on the effectiveness of our method", I do not agree that this is the case for the selection criterion in particular. The selection criterion can be completely decoupled from the ExcitationSolve ansatz, which can be used along with the typical gradient-based selection of ADAPT-VQE as an optimizer that doesn't affect the overall structure of the algorithm. I would like to see how ExcitationSolve behaves with this selection criterion, which would help understand how much of the improvement is coming from the change in selection criterion and how much is coming from the actual optimization.

The fact that "the optimal ordering of operators is an NP-hard problem in itself" is not relevant here. In fact, we could choose the best operator at each step by independently adding and optimizing each pool operator, and the total cost would be polynomial. The goal of selection heuristics is to bring us close to this optimal choice. It is possible to compare the operator selection method by testing the evolution of the algorithm with the two alternatives, and finding if there is a relevant difference in the number of parameters/operators of the final ansatz.

- Regarding parameter shuffling, I appreciate the inclusion of the sentence in the Supp. Information where the authors note that they did not find a systematic way of deciding when to apply it to avoid local minima. I think this is an important remark, as it is likely the most important direction for future work. Considering that, I think this should be addressed more clearly in the discussion when discussing local minima and future work.

Reviewer #3

(Remarks to the Author)

All my concerns were addressed in appropriate way (either in the manuscript, or if sufficient as part of the response).

Reviewer #4

(Remarks to the Author)

I co-reviewed this manuscript with one of the reviewers who provided the listed reports. This is part of the Communications Physics initiative to facilitate training in peer review and to provide appropriate recognition for Early Career Researchers who co-review manuscripts.

Version 2:

Reviewer comments:

Reviewer #2

(Remarks to the Author)

I appreciate the addition of Appendix B3, which added relevant information. However, as the authors state, the benefit of the modified selection criterion even when standard optimization is used (green curve) seems to stem from the initialization rather than the operator selection. To isolate the impact of the operator selection itself, I would suggest adding plots of energy error vs number of operators in the ansatz, instead of only including plots of energy error vs energy evaluations.

Response to review on Communications Physics manuscript COMMSPHYS-25-0627A
(Title: Fast gradient-free optimization of excitations in variational quantum eigensolvers)

We would like to thank the referees and editors for their time. We have addressed all comments of the referees. Below please find our response to each comment of each referee, including a summary of revisions made (referencing the specific locations in the revised manuscript). In addition to the revised manuscript file, we have also provided a version which highlights the revisions made in the manuscript (note that line and section numbers might differ from the ones in the plain version of the revised manuscript).

Reviewer #1 (Remarks to the Author):

This manuscript proposes a gradient-free quantum optimization method tailored to an ansatz constructed from excitation operators. The authors name this method ExcitationSolve, and it can be viewed as an extension of the Greedy Gradient-Free algorithm developed by Piquemal et al. While Piquemal et al. applied their method to Pauli operators appearing in UCCSD, the present study applies the optimization directly to the excitation operators themselves. At first glance, the idea may appear incremental, but its implementation is well motivated within the framework of quantum chemistry, and the method demonstrates significantly improved performance compared to conventional approaches. This advantage is observed not only in ideal simulations but also in the presence of shot noise and on real quantum devices. In light of these promising results, the manuscript has potential for publication, provided that several concerns are addressed.

1. The authors present their approach as a new method; however, it is essentially a variant of the optimization techniques developed in Rotosolve and by Piquemal et al. It would be preferable to adopt a consistent naming scheme that clearly reflects its connection to the method of Piquemal et al., so that readers can readily recognize it as an extension rather than an entirely new approach.

>> We thank the referee for this comment. As stated in the abstract, introduction, and results section, where the algorithm is defined, ExcitationSolve is an extension of Rotosolve when applied to fixed ansatzes. Therefore, we adopted a naming scheme consistent with Rotosolve that highlights the excitation-based operators central to our approach. We also made the connection to the work of Piquemal and colleagues explicit when introducing ExcitationSolve. To further acknowledge this prior work, we now mention their algorithm's name, Greedy Gradient-free Adaptive VQE (GGA-VQE), in both the introduction already and the results section when considering the adaptive ansatz context (see lines 86-88). While GGA-VQE uses the same operator selection strategy, it differs in that it does not re-optimize previously selected parameters, keeping them fixed instead. In contrast, ExcitationSolve employs Rotosolve-like optimization to update all parameters after each operator addition. This difference further justifies our emphasis on the Rotosolve connection in the naming scheme. We acknowledge that the name leans more toward Rotosolve than GGA-VQE. To balance this, we clearly state that ExcitationSolve for ADAPT-VQE is an extension of GGA-VQE and dedicate further details to it in lines 202-205. Overall, we prefer to retain the name ExcitationSolve because we believe it emphasizes the context of excitation-based operators and the primary

connection to existing optimization methods.

2. The authors claim that “the average estimate variance becomes inversely proportional to the total shot budget and, importantly, independent of the number of evaluation points.” However, prior work by Endo et al. demonstrates that other gradient-free methods “such as Free-Axis Selection (Fraxis), Free Quaternion Selection (FQS), and optimal parameter combinations” can achieve maximal optimization for a single gate without relying on gradients. Based on these findings, Endo et al. [58] conclude that allocating more shots to a single parameter configuration is preferable. The authors clearly should discuss this inconsistency.

>> We thank the referee to point out this seeming contradiction. We want to emphasize that our analysis is kept simple and focuses solely on the average statistical error over multiple energy evaluation estimates. This average error remains unchanged when the varying number of estimates this average is taken over while fixing total shot budget, as shot noise is additive. Hence, our analysis is limited to this average error perspective and is not tied to the resulting reconstruction quality in a more sophisticated manner. In particular, we do not address the propagation of the statistical (shot) error through the reconstruction to points further away from the sampling positions and neither the influence of the number and positioning of the sampling points on this error propagation. Moreover, we assume a constant observable variance $(\Delta H)^2$ across all parameter settings - a simplification that is generally not satisfied in practice. To acknowledge these limitations and guide readers to a more rigorous treatment of these issues, we have updated the manuscript and explicitly refer the reader to the work by Endo et al. (lines 1193-1200).

3. The authors state that “this approach can make ExcitationSolve more robust against noise” and that “least-squares estimation yields the optimal result under a normally distributed noise assumption” which is approximately fulfilled for pure states. However, while the proposed optimizations may be locally optimal, this does not necessarily guarantee convergence to a global minimum. To convincingly demonstrate the practical utility of these gradient-free local optimization methods, it is important to accumulate and present a range of empirical evidence. In fact, similar observations have already been reported in related methods such as Rotosolve/NFT, Free-Axis Selection and Free Quaternion Selection. We recommend that the authors emphasize the practical relevance of their approach by referring to these prior works.

>> We thank the referee for the suggestion. We extended the references that were listed in the appendix on related methods. This includes more references for the Free-Axis Selection, Free Quaternion Selection and Unitary Block Optimization Scheme (UBOS). More importantly, we followed the referee's suggestion to accumulate these works to provide empirical evidence for the general utility of this type of gradient-free local optimization approach and present this in the introduction (lines 73-77).

To further clarify that we do not claim that our optimizer would have any guarantee to converge to a global optimum, we added this statement to the discussion section (lines 477f). Nevertheless, we never intended to make such claims, and the excerpts cited by the referee are in the context of determining the coefficients in the analytic form under shot noise, i.e., reconstructing the energy function in a single parameter optimally for certain assumptions on the noise distribution. They are not related to finding the global optimum of the energy function in all parameters.

4. In Fig. 6, which shows noiseless simulations using a fixed-depth UCCSD ansatz, the energy sometimes increases even though analytical minimization is used in each ADAPT step. If the optimizer is truly analytical, the energy should not increase under ideal conditions. This behavior seems inconsistent and should be explained.

>> We thank the referee for pointing this out. The observation is valid in the case of H2 and H3+ at around 5 and 170 energy evaluations, respectively. We attribute these energy increases to numerical inaccuracies since at both these energy increases we already achieved an absolute energy difference to the FCI energy smaller than $1e-13$. We added a clarification to the manuscript (lines 296-298).

5. In addition, the authors discuss the method's robustness to shot noise; however, this does not necessarily extend to noise on real quantum devices. In the real-device experiment shown in Figure 9, the authors report that the energy estimated directly from the circuit is higher than that of the Hartree-Fock state, yet the optimal parameters themselves can still be identified. This appears to be intuitively contradictory. If this observation is accurate, does it imply that real-device noise introduces systematic errors? If not, would this favorable behavior persist as the system size increases? The authors should address this issue in more detail.

>> We thank the referee for pointing out this intriguing observation and to spark a more detailed discussion (based on the questions raised in this remark). The referee's interpretation in the the first question aligns with our experience. We elaborated on this further in the manuscript (lines 402-407). To address the second question: as pointed out in the remark, our (theoretical) discussion of the method's noise robustness is mostly limited to shot noise, which is more structured (e.g., modeled faithfully by normally distributed noise for practical shot numbers, cf. lines 1165f). Furthermore, we limited the real-hardware experiments to small molecules as the energy estimates for larger molecules were too noisy and unreliable, due to the significantly increased circuit depth for multiple excitation operators. Therefore, we suspect that this (favorable) behavior does not persist as the system size increases as added in (lines 410-413).

Reviewer #2 (Remarks to the Author):

The authors propose a new optimization method, termed ExcitationSolve, aimed at optimizing ansätze for the electronic structure problem. In particular, the optimizer uses the structure of fermionic (or qubit) excitations to exactly minimize one-dimensional cost landscapes based on just a few energy evaluations per parameter. This generalizes the well-known RotoSolve algorithm, which applies to ansätze composed of unitaries generated by individual Pauli strings. The authors compare their protocol against optimization methods in the literature and find that it significantly reduces the number of function evaluations required to reach a given accuracy. Based on hardware experiments on up to 6 qubits, the protocol is shown to also be more resilient to noise. I believe that the work is relevant and could recommend publication once the remarks below are addressed.

- Following the introduction of ADAPT-VQE, the authors claim that "the VQE optimization problem is generally challenging" and that there is a "large number of local minima" which leads optimizers to fail. However, this is the opposite of what we observe in ADAPT-VQE landscapes - the parameter initialization strategy seems to effortlessly lead to good quality minima that are easily found by local optimizers. See <https://www.nature.com/articles/s41534-023-00681-0>

>> We thank the referee for bringing up this ambiguity. In fact, the cited text in the remark was not meant to be connected to ADAPT-VQE. We moved the brief introduction of ADAPT-VQE to the end of the second paragraph (Introduction section) instead to increase the clarity.

- When the authors state that each parameter must occur only once in the ansatz for Eq. 3 to hold, they should specify that they mean that the excitation must only occur once in the ansatz with this parameter. Decomposing an excitation into basic gates will in general require more than one single-qubit rotation gate with this rotation angle, but I believe this will not hinder the application of the formula, as we can see the implementation of the unitary as a block in the circuit, regardless of the actual physical implementation.

>> This is correct. Not only does that mean that the excitation must only occur once in the ansatz with this parameter, but also that this parameter can not be used by any other excitation. We have addressed the remark (lines 140-143) by specifying the number of occurrences in terms of "parameterized excitations" to avoid confusion. The application of our formula is not hindered by the fact that a decomposition into basic gates introduces multiple rotation gates sharing the same angle. In Sec. 2.1 (lines 176-180) we discuss that the order of the Fourier series is exponentially overestimated if one considers the decomposition in terms of, e.g., Pauli rotations.

- The description of the algorithm in the main text and in the scheme of Fig. 2 is a bit confusing. The authors use k to label the iteration and j to label the operator whose coefficient is modified in this iteration, but they do not specify which j is optimized in each iteration. From the supplementary material, it seems that parameters are updated in order, with each individual update counting as one iteration, and the optimization continues cyclically until a certain convergence criterion is met. This should be clear from the main text and the flowchart in Fig. 2.

>> We thank the referee for raising this point. It is correct that parameters can be updated in any order, which was described in the main text before by "The order in which the parameters \hat{I}_j are optimized can be chosen freely". However, we agree that further clarification is necessary, especially due to the two different index labels k and j . We expanded the main text (line 129), flowchart caption (Fig. 2) and appendix (lines 1134f) accordingly.

- The variable N that appears in the convergence criterion does not seem to be defined.

>> We thank the referee for pointing this out. Even though the variable N is defined in the introduction as the total number of parameters, we understand that recalling the definition in Figure 2 (caption) can be of benefit and increase clarity, specifically to understand the convergence criterion. We also changed an unintentional redefinition of N in the Methods sections to a different letter to ensure that N consistently

associated with the number of parameters throughout the work.

- The authors focus the discussion on page 6 on occupied-to-virtual excitations, but it's worth mentioning generalized excitations since the algorithm readily applies to those also.

>> We thank the referee for pointing this out. We have added a note at the end of Sec. 2.1 (lines 188-189) to mention generalized excitations.

- The use of the expression "globalized ADAPT-VQE operator selection criterion" might confuse the readers, since the word "globalized" might suggest that this criterion takes into account the interplay between the parameters in a global optimization, which is not the case. Similarly, the sentence "We select the operator that achieves the strongest decrease in energy to be appended to the ansatz" is misleading, since the strongest decrease in energy mentioned here is the one achievable by changing only one variational parameter. The strongest energy decrease that might be produced by adding an operator and performing a full optimization remains unknown. I would suggest making this clear. While this is discussed later, even just rephrasing this as "that achieves the strongest decrease in energy when appended to the previous ansatz with a fixed coefficient" might avoid misconceptions.

>> We thank the referee for this remark. We agree that (a) the expression "globalized" might confuse the reader. However, we believe that the explanations and definitions provided in the main text sufficiently clarify the intended meaning. The same applies to (b) the expression "strongest decrease". Nevertheless, we appreciate the suggestion and have taken steps to improve the clarity of the manuscript by slightly adjusting the wording and adding further explanations.

(a) we replaced the term "globalized" with "globally-informed", which is consistent with the terminology previously introduced in the context of ExcitationSolve for the fixed-ansatz VQE. In that context, the parameters are optimized individually based on a reconstruction over the full (global) domain of the single parameter, and we believe this should make the intended meaning clear to the reader and prevent confusion with global optimization. Furthermore, we would like to point out that, to the best of our knowledge, there is no established or commonly used terminology for this type of globally-informed selection criterion or heuristic. That said, we do acknowledge that in the optimization (mathematics) literature, ExcitationSolve in the fixed-ansatz variant corresponds to a coordinate descent algorithm, as already noted in the previous version of the manuscript. Additionally, we now also mention that the optimization of a single parameter to its global minimum in each iteration is referred to as an "exact line search" in this context, and we have added this clarification to lines 146-150.

(b) we changed the phrase "strongest decrease" to "strongest immediate decrease" at its first occurrence (line 197), to emphasize that this is not related to a global optimization. To avoid verbosity, and given that the referee points out the concept is explained unambiguously in the following text, we refrain from repeating the full phrase "strongest immediate decrease" and instead keep the shorter "strongest decrease" in subsequent occurrences.

- In Fig. 4, it is unclear whether the example provided corresponds to an actual scenario observed in numerical simulations or if it is fabricated to showcase the motivation - could the authors make this clear?

>> We have now made clear that the example provided is artificially

fabricated by considering two different excitations on a minimalist random three-qubit Hamiltonian (lines 216-218).

- I believe that the selection criterion deserves more attention. The authors claim that their method is superior, but it is difficult to assess this when the selection criterion is implemented along with the optimization method. Comparing ADAPT-VQE with the gradient vs energy-based selection criterion using the same optimizer would be ideal.

>> We thank the referee for pointing this out. We agree that studying the effects independently would lead to a more direct comparison. However, we believe that the provided experiments and extensive analyses provides convincing evidence for our conclusions on the effectiveness of our method. As mentioned in the text, there are two benefits to the operator selection: one is the reduced number of parameters and the second is the initialization strategy. As the optimal ordering of operators is an NP-hard problem in itself it is unknown what the best order for operators would look like. Therefore it is not sensible to compare the operator selection before convergence is reached and to judge from the overall result which choice of operators is best. We judge this by the number of operators that were necessary to reach convergence, for which ExcitationSolve outperforms the original ADAPT for LiH and H2O and ties for H2 and H3+. Those numbers are reported in the main text (lines 328-330).

The second benefit becomes apparent when looking at Fig. 11 in the appendix. Here we can see that through this initialization strategy, ExcitationSolve already achieves most of the energy reduction already when appending the operators to the ansatz, without ever re-evaluating previously appended operators (bottom row of each plot). And indeed there is no visible benefit beyond a single re-evaluation, because the single operator optimum seems to be a good approximation to the global minimum. This last observation however is only a heuristic and could be different for cases where operators are strongly coupled and their common minimum is far away from the individual optima.

Regarding both benefits, an analysis and interpretation of Fig. 11 was provided in appendix B2.

- In Fig. 5, the authors should include the scale of energy/error values associated with the colors. The final 1D and 2D errors should also be included to show the magnitude of the benefit of using the 2D optimization.

>> As for Fig. 4, we have added a remark (251-252) that the example is fabricated. Having clarified that both examples serve the purpose of showcasing a motivation based on fabricated examples rather than an actual observed scenario, we believe that including an explicit scale does not provide any merit to the reader here.

- The authors should specify the order of the excitations they consider for the UCCSD ansatz.

>> We thank the referee for pointing out this important detail. We added a sentence to the main text (lines 280-282) clarifying that we use the UCCSD ansatz in the first-order Trotter approximation, and that we first apply all double excitation, then all single excitations. Due to the high number of excitations, we omit an explicit specification of the total order of the excitations in the ansatz. However, to make our experiments more clear and reproducible, we added a statement to the appendix indicating the Pennylane version that was used to generate the excitation order (line 882).

- More details should be provided about the basis set considered for the molecules. It would also be helpful to mention the number of qubits used to represent each in the main text in addition to the plots.

>> We thank the referee for pointing this out. We added a clarification in the main text that we use the STO-3G basis and added the number of qubits in the main text (lines 283-284).

- Systems are restricted to be small and/or weakly correlated, in particular because bond distances are set to equilibrium. It would be interesting to test stretched bond distances and slightly larger, strongly correlated molecules such as H6. I appreciate that the authors include bond dissociation curve plots for the UCCSD ansatz, but (1) they do not consider the same for ADAPT-VQE, (2) they do not include error plots, and (3) they do not consider a staple difficult system such as H6. In particular, H6 at stretched bond distances is known to lead to a roadblock in ADAPT-VQE known as gradient troughs (see <https://www.nature.com/articles/s41534-023-00681-0>). It would be interesting to understand whether ExcitationSolve could help mitigate this problem.

>> We thank the reviewers for this interesting remark. In response we have investigated ExcitationSolve on H6 at stretched bond distances. We find that we can get a speed-up in terms of energy evaluations of a factor 5 when using ExcitationSolve, however an equal number of operators are required to reach convergence in both cases. ExcitationSolve therefore does not solve the roadblock here. We conjecture that the reason behind this lies in the enhanced electron correlation that results from the stretched bond. To accurately depict the ground state of such a system many operators are required, independent of the optimizer that is used to append and optimize them.

- The authors plot the energy error against the number of energy evaluations, but it is not clear how they factor the gradient evaluations into the costs for the case of gradient-based optimizers.

>> We thank the referee for requesting further clarification. The four-term parameter-shift rule is employed and matches excitation operators and requires four evaluations. We extended the main text to clarify this point in (lines 271f). (See details for gradient-based optimizer convergence curves matching the vertical shadings in our comment to the following remark.)

- In Fig. 6, it is unclear why methods such as BFGS and GD follow this pattern, where they are constant across one vertical shading. According to the text, the shading marks one iteration over all parameters - what does this mean? One iteration will require a different number of function evaluations depending on the method, so it is strange that several optimizers have a constant energy value across these shadings.

>> We thank the referee for pointing this out. This is indeed a good observation and is connected to the matching number of energy evaluations required for the parameter-shift rule to compute the gradients in these methods. In addition to the extensions based on the previous remark, we added further clarification to the caption of Fig. 6. The BFGS optimizer needs additional energy evaluations per update-step to approximate the Hessian. Therefore the BFGS updates do not happen immediately after each vertical shading.

- When comparing the number of function evaluations required across the bond dissociation curve in Fig. 8, it would be relevant to include the

final VQE error in addition to the HF error.

>> We thank the referee for pointing this out. We added a Appendix section (B.3) where we present the final VQE errors. We refer to this Appendix in the main text (lines 347-348). We think that adding additional plots to Fig. 8 showing the final VQE error per bond distance would make Fig. 8 to crowded.

- In Table I, I suppose that the "theoretical" vs "effective" number of energy evaluations required to reconstruct the function comes from the fact that the energy value for the unshifted coefficient can be recycled from previous calculations, but it might be worth clarifying and referring to earlier statements concerning the subject.

>> We thank the referee for asking for further clarification on that matter. Yes, this interpretation of "theoretical" vs "effective" is correct. We still added further clarification in the caption of Table 1.

- In table I, the authors should note that while general parameter-shift rules for these generators require 4 circuit evaluations per parameter, this might be reduced to 2 if the wavefunction is real, which is the case here. Further, writing the cost of the parameter shift rules in terms of the number of energy evaluations is not accurate, because the excitation generators have three eigenvalues. This difference in the spectrum leads the PSR formulas to require circuits that differ from the ones necessary to evaluate the energy. This is discussed in Ref. 35.

>> We had previously intentionally spared the details about this two-shift method in the main text (yet provided in the appendix) since, as the reviewer rightly pointed out, the required circuits are not the same as for energy evaluations.

Given that this might be misleading concerning the efficiency and potential of gradient-based approaches, we have now inserted additional information about this method in the text (lines 423-426), appendix (lines 1347-1351) as well as Table I. In the table, we now talk about the cost in terms of shift to make the terminology consistent with the two-shift method for real-valued wave functions.

We highly appreciate this remark!

- The authors refer to their experiments as being performed "on a 127-qubit IBM quantum device", which is misleading considering that the experiments only use up to 6 qubits. The full size of the device is not relevant for the experiments performed.

>> We thank the referee for pointing this out. We removed the "127-qubit" description in contexts where it seemed misleading.

- Parameter shuffling seems necessary to achieve convergence in the largest molecule, but I am concerned that there is no systematic way of deciding when to apply this. The manuscript does not seem to include enough details about when to apply shuffling or what exactly this consists of.

>> We thank the referee for pointing out this important detail. We added additional information about the shuffling in the respective Appendix section, now B.4 (lines 1073-1079).

- Most of the Methods section ("ExcitationSolve for multiple occurrences of parameters", "Classical minimization of analytic energy reconstructions", "Reconstruction strategies for noise robustness") could be moved into the supplementary material, since it mostly consists of technical details which are peripheral to the results and the manuscript could benefit from being

more succinct.

>> We thank the referee for the valuable suggestion. We reevaluated the positioning of the relevant sections in relation to their importance in the main text. To improve conciseness, we moved several sections and parts to the supplementary material/appendix (Sec. C, "ExcitationSolve algorithmic details"). The section "Reconstruction strategies for noise robustness" was moved in its entirety because it mainly contains algorithmic details that were not actively used in the presented experiments. Parts of "ExcitationSolve for multiple occurrences of parameters" were also moved to the supplementary material/appendix. In particular, the most generic energy reconstruction expression (formerly Eq. (10)) now constitutes appendix section C.3, "ExcitationSolve for multiple occurrences of multiple parameters." However, due to its relevance, particularly for higher-order Trotterization (multi-layer UCC ansätze), as mentioned in the main text, we kept a brief explanation of ExcitationSolve for a single parameter occurring multiple times. This explanation is relevant in quantum chemistry applications. We left section "Classical minimization of analytic energy reconstructions" in the main text methods section because it is used directly in the ExcitationSolve implementation and the experiments and results presented in the main text.

Reviewer #3 (Remarks to the Author):

In the manuscript, authors present ExcitationSolve, an algorithm that is generalization of Rotosolve algorithm for gates of the form $\exp(iGt)$ for generators G satisfying $G^3 = G$. The presented algorithm was applied for parameter optimization of ansätze, pool gate selection in ADAPT-VQE, as well as extends the existing methods defined for multiple occurrences of single parameters in the ansatz. While at first this might seem to be incremental improvement, there are many important generators that satisfied the above equation including excitation-annihilation operators or a fermionic-inspired gates like QEB or CEO, which makes the research line timely and relevant for current community interest.

I have no doubt about quality and importance of the pursued research and they alone convince me recommend the paper for publication. However, at the same time I would like to request a numerous (mostly small) changes to the article that would improve its readability.

I would like to comment on the terminology introduced by the researchers:

1. frequent used of "globally-informed" w/o or with confusing interpretation. From the ansatz perspective, ExcitationSolve is embarrassingly local algorithm, however it is global from the perspective of single-parameter landscape, where it "immediately" jumps into global minimum. If the latter context is intended, in my opinion, the terminology is a bit overused, but it is fine as long as authors will define this term properly. This and similar terms are used frequently throughout the paper

>> We thank the referee for the comment. Although the term "globally-informed" is used frequently throughout the manuscript, we consistently use it to refer to the evaluation of the energy landscape over

the full domain of a single variational parameter (i.e., global in the single-parameter landscape). However, we believe the explanations and definitions in the main text sufficiently clarify the intended meaning.

For further clarity, we note that the connection to coordinate descent was made in the previous version of the manuscript. In this context, we improved the explanation by adding a connection to the well-established optimization literature term "exact line search" (lines 146-150) to describe the process of minimizing along the single-coordinate descent direction. We hope this additional clarification helps to avoid potential misinterpretations. Furthermore, to make the terminology used in the paper more consistent, we changed the usage of "globalized" for the operator selection criterion in ExcitationSolve in ADAPT-VQE to "globally-informed."

2. quantum-aware is not defined

>> We thank the referee for catching the lack of clarity and definition of the terminology here. We adjusted the introduction of quantum-aware optimization in the introduction to make the definition clear and highlight the contrast to other optimizers. (lines 58-61)

In addition I would like to point out some other a bit confusing to my understanding terminology or notation used

1. 144: occupied/virtual orbitals " this terminology is inherited to my understanding from quantum chemistry, and it loses its meaning when Hermitian conjugate part in Eq. (4) is present, in which case they would have the opposite meaning. I would avoid using this term. In addition note that such operators can be acting solely on primarily occupied (virtual) orbitals for sufficiently advanced state.

>> From our understanding, the sets of occupied and virtual orbitals simply characterize the orbitals based on whether they are occupied or not in the approximate ground state obtained from Hartree-Fock theory.

Indeed, the Hermitian conjugate has the opposite effect as it moves electrons from virtual to occupied orbitals, but we do not view this to be conflicting with the prior terminology, since this only occurs if previously an electron has been moved to this virtual orbital (and this electron must have originated from an occupied orbital). Given that many other works in the realm of quantum computing for quantum chemistry use the same terminology, we would like to avoid changes here.

The additional note that such operators may act solely on the primarily occupied (virtual) orbitals aligns with the remark of reviewer 2 that generalized excitations should be mentioned. We appreciate the hint and have inserted a note (lines 188-189).

2. 244: It would be more accurate to use "chemical precision" instead of "accuracy" since the latter is more connected to final, infinite-dimension solution of Schrodinger equation. While in 244 this could be accepted the way it is introduced, term "accuracy" is misused in e.g. Fig. 6. Due to the nature of the manuscript, I believe replacing accuracy with precision everywhere in the paper might be optimal choice

>> We thank the referee for pointing this out. We agree that, from a technical standpoint, "chemical precision" may be more accurate terminology. However, "chemical accuracy" is a well-established term in the quantum computing for quantum chemistry literature in this context - used, for example, in the original VQE paper (Ref. [1]). While we acknowledge the

distinction, we believe that using "chemical accuracy" aligns with the conventions of the field and is unlikely to cause confusion among readers familiar with the topic. In light of the referee's comment, we took the opportunity to revisit the literature and have added a clarifying remark and relevant references (lines 267f) to address the terminology more explicitly.

3. 328: Parameter shuffling is not explained

>> We thank the referee for pointing this out. We added additional information about the shuffling in the respective Appendix section, now B.4 (lines 1073-1079).

I would also be interested in having some more discussion:

1. Eq. (7) and below: I believe it would be interesting to comment difference between Rotosolve and ExcitationSolve in this context to strengthen the improvement of the latter: note that Rotosolve solves simpler case where local minima are global minima

>> We thank the referee for this hint on how to strengthen the improvement our method provides over previous works. We added a statement in lines 230-232.

2. 962-965: I don't understand why larger rotation angles would increase hardware noise given that A) eventually rotations are periodic so the rotation time can be moved into some fixed interval B) IBM machines have virtual Z rotations

>> We thank the referee for pointing this out. Indeed, the excitations are generally 2π periodic, however, (if the coefficients for the lower frequency component vanish) can effectively be only π periodic. Hence, remarked point A), i.e., moving points into the period interval, is an equivalent interpretation of our approach of choosing the minimum with the smaller angle in such cases. Regarding remarked point B), we agree that the angle magnitude does not directly affect the error, as IBM implements rotations using virtual Z rotations. We thank the referee for this valuable note. However, since we filter out operators with parameter values close to zero during transpilation, it remains important to avoid using larger but periodically equivalent rotations. Doing so allows us to bypass the transpilation of the entire excitation operator (not only the Z-rotations), leading to shallower, less erroneous circuits. We corrected our statement in the text (lines 992-996).

Finally, I would appreciate if authors would improve readability of some parts in Appendices:

1. A.3, Table 3.: What is the meaning of values under VQE columns? Do they correspond to stopping conditions for parameter optimization?

>> That is correct. We have changed the heading to "parameter optimization" to clarify and thank the referee for this improvement.

2. 876-890: I'm a bit confused with explanation on why only 16 evaluations are needed

>> We have reformulated that paragraph, hopefully making it clearer (lines 898-905). It now reads: "For ExcSolve2D, the two operators that have the biggest impact are selected, optimized using 2D ExcitationSolve and then appended to the ansatz. As we already calculated the energies for the individual operators in the ranking, we can reuse those in the

optimization. The five single parameter values can be viewed as the shifts of θ_i where $\theta_j=0$ and vice versa. The value for $\theta_i=\theta_j=0$ appears in both selections, so only nine values can be recycled. This leaves $25-9=16$ energy evaluations to be done. So overall we calculate the energies for 24 sets of (θ_i, θ_j) , but the effort is split between operator ranking and actual optimization of the chosen operators."

3. 909: for $\hat{I}_j = 0$ is energy evaluation repeated, or reused from previous iteration?

>> We thank the referee for pointing this out. We clarified our approach, by explaining that we repeat (not reuse) the energy evaluation (lines 935-936).

4. 910: is parameter shift rule used for gradient-based methods?

>> We thank the referee for pointing this out. We clarified our approach by explaining that we use the parameter-shift rule for gradient-based optimizers (lines 939-940).

5. 934: $1/8$ seems to be pretty large threshold. Didn't it have significant negative effect on evaluation quality

>> We thank the referee for raising the concerns. While $1/8$ may appear like a large threshold, it is important to consider that the unit is radians. Hence, given the 2π periodicity, the interval induced by the thresholds $[-1/8, 1/8]=[-0.04\pi, 0.04\pi]$ marks not more than 4% of the entire period 2π . Relevant parameters (corresponding to excitations that are involved in the ground state) in H_2 and H_3^+ never fall into this threshold range, which is why no negative effect on the evaluation quality is observed. On the other hand, choosing this threshold too small, i.e., transpiling more excitation operators than necessary, leads to a significantly worse accuracy in the energy estimates due to the increased depth of the circuits and hence substantial hardware noise. We added the justification to the appendix (lines 961f).

6. Fig. 11. could have more informative labels, in particular I believe consecutive rows correspond to consecutive molecules

>> We have added labels to the rows of the figure to mark the different molecules and thank the referee for pointing out this improvement.

7. Eq. (26): I checked derivations of formulas, but I'm unsure about the number 3 next to the last term. Could the authors make the derivation explicit (e.g. in the response?). In case this is incorrect, similar fix is likely required in Eq. (44).

>> To resolve this doubt, it is best to start from Eq. (25). Here, the last term this comments concerns with appears multiple times. Importantly, also with a prefactor of \cos^2 . Expanding this as $\cos^2(x) = (1+\cos(2x))/2$ gives rise to the additional constant of $/2$, which is why in the end we have $3/2$ in Eq. (26).

8. C.2.: While I do agree with the proved statement, kindly rewrite the proof using concise product notation $\prod_{i=1}^m (\hat{A}_i)$. Please note that a convention whether product is taken left-to-right or right-to-left should be defined. In particular, I believe there is mistake in the last line of Eq. (35) (superfluous v_1 ?)

>> We have rewritten the proof (now in D.2) using product notation.

In the same manner, we have adjusted Eq. (4) in the main text to also use product notation, including a remark on the product order. We believe that the proof is now more comprehensive to read and therefore thank the reviewer for this suggestions.

Reviewer #4 (Remarks to the Author):

Reviewer #4 co-reviewed this manuscript and their remarks are part of one of the other reviews.

Response to review on Communications Physics manuscript COMMSPHYS-25-0627B
(Title: Fast gradient-free optimization of excitations in variational quantum eigensolvers)

We want to thank the referees and editor for their time. We have addressed all the comments from referee #2, while all other referees recommended accepting the manuscript without further modifications. Below, please find our response to each comment of referee #2, including a summary of revisions made (referencing the specific locations in the revised manuscript).

In addition to the revised manuscript file, we have also provided a version which highlights the revisions made in the manuscript (note that line and section numbers might differ from the ones in the plain version of the revised manuscript).

Reviewer #2 (Remarks to the Author):

I appreciate the improvements to the manuscript, which addressed many of the questions I had. Once the two points below have been addressed, I can recommend publication.

- While I agree with the authors' statement that "extensive analyses provides convincing evidence for our conclusions on the effectiveness of our method", I do not agree that this is the case for the selection criterion in particular. The selection criterion can be completely decoupled from the ExcitationSolve ansatz, which can be used along with the typical gradient-based selection of ADAPT-VQE as an optimizer that doesn't affect the overall structure of the algorithm. I would like to see how ExcitationSolve behaves with this selection criterion, which would help understand how much of the improvement is coming from the change in selection criterion and how much is coming from the actual optimization. The fact that "the optimal ordering of operators is an NP-hard problem in itself" is not relevant here. In fact, we could choose the best operator at each step by independently adding and optimizing each pool operator, and the total cost would be polynomial. The goal of selection heuristics is to bring us close to this optimal choice. It is possible to compare the operator selection method by testing the evolution of the algorithm with the two alternatives, and finding if there is a relevant difference in the number of parameters/operators of the final ansatz.

>> We thank the referee for suggesting further analysis of the effectiveness of the proposed selection criterion. In order to assess the utility which stems purely from our operation selection criterion, we have, as suggested, conducted simulations for LiH and H2O using ExcitationSolve-based optimization and gradient-based selection. For the sake of completeness, we further conducted simulations for both systems using the opposite approach, that is gradient-based optimization (GD) with energy-based selection (using ExcitationSolve).

We find that using ExcitationSolve as either optimizer or selection criterion is already much faster than purely gradient-based approaches.

We find that indeed, using the energy-criterion leads to ansätze consisting of fewer operators regardless of the chosen parameter optimizer, providing strong evidence for the improved quality in the operator selection over the local gradient-based criterion. Even when GD is used as parameter optimizer the optimization requires fewer resources, mainly due

to the initialization strategy included in the ExcitationSolve operator selection. Interestingly, we see that using gradient-based selection with ExcitationSolve as optimizer performs even faster than the opposite approach, despite requiring more operators overall.

Therefore, using ExcitationSolve for both the optimization and selection joins the best out of both approaches.

For a more in-depth discussion of these findings, we have added lines 338-341 to the main text, and present our new findings in Appendix B.3.

- Regarding parameter shuffling, I appreciate the inclusion of the sentence in the Supp. Information where the authors note that they did not find a systematic way of deciding when to apply it to avoid local minima. I think this is an important remark, as it is likely the most important direction for future work. Considering that, I think this should be addressed more clearly in the discussion when discussing local minima and future work.

>> We thank the referee for this suggestion. We agree and added a statement to the Discussion section (lines 481-487) addressing this aspect of future work.

Response to review on Communications Physics manuscript COMMSPHYS-25-0627C
(Title: Fast gradient-free optimization of excitations in variational quantum eigensolvers)

We want to thank referee #2 and the editor for their time. Below, please find our response to the comment of referee #2.

Furthermore, we have addressed all points in the checklist of editorial requests and submitted all requested files along with the final version of the manuscript.

Reviewer #2 (Remarks to the Author):

I appreciate the addition of Appendix B3, which added relevant information. However, as the authors state, the benefit of the modified selection criterion even when standard optimization is used (green curve) seems to stem from the initialization rather than the operator selection. To isolate the impact of the operator selection itself, I would suggest adding plots of energy error vs number of operators in the ansatz, instead of only including plots of energy error vs energy evaluations.

>> We thank the referee for suggesting to improve the presentation of the effectiveness of ExcitationSolve through an alternative plot. It is true that the plot added in the previous revision (Fig. S3, Supplementary Information) emphasizes on performance gains in terms of energy evaluations (x-axis) resulting from the initialization. However, even when comparing ExcitationSolve (red curve) with the variant where the ADAPT gradient selection criterion is used (green), we can see that the former achieves lower energies with the same number of operators added. For this, one can compare the end of the regions of markers with lighter colors. More importantly, to make this information directly apparent, we added a table in the previous revision already (Table S4, Supplementary Information), which compares the exact number of operators in the ansatz for both molecules and each optimizer combination at the two relevant stages: when chemical accuracy was reached and upon full convergence. As interpreted in the accompanying text, we can see that the ExcitationSolve selection leads consistently to a lower number of operators for full convergence (the same and lower for chemical accuracy for LiH and H₂O, respectively.) Therefore, we are confident that the provided data added in the previous revision in form of the plot and table suffice to demonstrate that the benefit of the ExcitationSolve operator selection stems not only from the initialization but also from the actual selection criterion.

In the manuscript, authors present ExcitationSolve, an algorithm that is generalization of Rotosolve algorithm for gates of the form $\exp(iGt)$ for generators G satisfying $G^3 = G$. The presented algorithm was applied for parameter optimization of ansatze, pool gate selection in ADAPT-VQE, as well as extends the existing methods defined for multiple occurrences of single parameters in the ansatz. While at first this might seem to be incremental improvement, there are many important generators that satisfied the above equation including excitation-annihilation operators or a fermionic-inspired gates like QEB or CEO, which makes the research line timely and relevant for current community interest.

I have no doubt about quality and importance of the pursued research and they alone convince make me recommend the paper for publication. However, at the same time I would like to request a numerous (mostly small) changes to the article that would improve it's readability.

I would like to comment on the terminology introduced by the researchers:

1. frequent used of 'globally-informed' w/o or with confusing interpretation. From the ansatz perspective, ExcitationSolve is embarrassingly local algorithm, however it is global from the perspective of single-parameter landscape, where it 'immediately' jumps into global minimum. If the latter context is intended, in my opinion, the terminology is a bit overused, but it is fine as long as authors will define this term properly. This and similar terms are used frequently throughout the paper
2. quantum-aware is not defined

In addition I would like to point out some other a bit confusing to my understanding terminology or notation used

1. 144: occupied/virtual orbitals – this terminology is inherited to my understanding from quantum chemistry, and it loses it's meaning when Hermitian conjugate part in Eq. (4) is present, in which case they would have "the opposite" meaning. I would avoid using this term. In addition note that such operators can be acting solely on primarily occupied (virtual) orbitals for sufficiently advanced state.
2. 244: It would be more accurate to use 'chemical precision' instead of 'accuracy' since the latter is more connected to final, infinite-dimension solution of Schrodinger equation. While in 244 this could be accepted

the way it is introduced, term ‘accuracy’ is misused in e.g. Fig. 6. Due to the nature of the manuscript, I believe replacing accuracy with precision everywhere in the paper might be optimal choice

3. 328: Parameter shuffling is not explained

I would also be interested in having some more discussion:

1. Eq. (7) and below: I believe it would be interesting to comment difference between Rotosolve and ExcitationSolve in this context to strengthen the improvement of the latter: note that Rotosolve solves simpler case where local minima are global minima
2. 962–965: I don’t understand why larger rotation angles would increase hardware noise given that A) eventually rotations are periodic so the rotation time can be moved into some fixed interval B) IBM machines have virtual Z rotations

Finally, I would appreciate if authors would improve readability of some parts in Appendices:

1. A.3, Table 3.: What is the meaning of values under VQE columns? Do they correspond to stopping conditions for parameter optimization?
2. 876–890: I’m a bit confused with explanation on why only 16 evaluations are needed
3. 909: for $\theta = 0$ is energy evaluation repeated, or reused from previous iteration?
4. 910: is parameter shift rule used for gradient-based methods?
5. 934: $\frac{1}{8}$ seems to be pretty large threshold. Didn’t it have significant negative effect on evaluation quality
6. Fig. 11. could have more informative labels, in particular I believe consecutive rows correspond to consecutive molecules
7. Eq. (26): I checked derivations of formulas, but I’m unsure about the number 3 next to the last term. Could the authors make the derivation explicit (e.g. in the response?). In case this is incorrect, similar fix is likely required in Eq. (44).

8. C.2.: While I do agree with the proved statement, kindly rewrite the proof using concise product notation $\prod_{i=1}^m(\cdot)$. Please note that a convention whether product is taken left-to-right or right-to-left should be defined. In particular, I believe there is mistake in the last line of Eq. (35) (superfluous $a_{v_1}^\dagger$?)